# Quantitative mapping of dense microtubule arrays in mammalian neurons

Eugene A Katrukha, Daphne Jurriens, Desiree M Salas Pastene, Lukas C Kapitein*

Cell Biology, Neurobiology and Biophysics, Department of Biology, Faculty of Science, Utrecht University, Utrecht, Netherlands

**Abstract** The neuronal microtubule cytoskeleton underlies the polarization and proper functioning of neurons, amongst others by providing tracks for motor proteins that drive intracellular transport. Different subsets of neuronal microtubules, varying in composition, stability, and motor preference, are known to exist, but the high density of microtubules has so far precluded mapping their relative abundance and three-dimensional organization. Here, we use different super-resolution techniques (STED, Expansion Microscopy) to explore the nanoscale organization of the neuronal microtubule network in rat hippocampal neurons. This revealed that in dendrites acetylated microtubules are enriched in the core of the dendritic shaft, while tyrosinated microtubules are enriched near the plasma membrane, thus forming a shell around the acetylated microtubules. Moreover, using a novel analysis pipeline we quantified the absolute number of acetylated and tyrosinated microtubules within dendrites and found that they account for 65–75% and ~20–30% of all microtubules, respectively, leaving only few microtubules that do not fall in either category. Because these different microtubule subtypes facilitate different motor proteins, these novel insights help to understand the spatial regulation of intracellular transport.

*For correspondence:
l.kapitein@uu.nl

**Competing interests:** The authors declare that no competing interests exist.

## Introduction

The extended and polarized morphology of neurons is established and maintained by the cytoskeleton (*Stiess and Bradke, 2011*; *Bentley and Banker, 2016*). One of the functions of the microtubule cytoskeleton is to provide a transport network inside the neurons long axon and branched dendrites (*Kapitein and Hoogenraad, 2015*; *Burute and Kapitein, 2019*). Directional transport is enabled by the structural polarity of microtubules, which is recognized by motor proteins that drive transport to either the minus end (dynein) or plus end (most kinesins). Using this network, intracellular cargos attached to microtubule-based motor proteins (kinesins and dynein) can be shipped between different neuronal compartments (*Hirokawa et al., 2010*; *Kapitein and Hoogenraad, 2011*). To facilitate proper delivery, transport is regulated at multiple levels. For example, molecular motors, motor adaptor proteins, and cargos themselves undergo tight biochemical regulation in response to changes in metabolic state and extracellular cues (*Hirokawa et al., 2010*; *Tempes et al., 2020*). An equally important component is the composition and spatial distribution of microtubule tracks, which is the main subject of this study.

Previous work has revealed that the affinity of motors for the microtubule lattice can be modulated by microtubule-associated proteins (MAPs) or by post-translational modifications (PTMs) of tubulin (*Atherton et al., 2013*; *Sirajuddin et al., 2014*; *Jurriens et al., 2021*; *Monroy et al., 2020*; *Park and Roll-Mecak, 2018*; *Janke and Magiera, 2020*). For example, kinesin-1 motors move preferentially on microtubules marked by acetylation and detyrosination (*Cai et al., 2009*; *Dunn et al., 2008*), while kinesin-3 prefers tyrosinated microtubules (*Guardia et al., 2016*; *Tas et al., 2017*; *Lipka et al., 2016*). When tubulin is incorporated into the microtubule lattice, it carries a genetically encoded C-terminal tyrosine, which can subsequently be proteolytically removed to yield detyrosinated microtubules (*Kapitein and Hoogenraad, 2015*; *Janke and Magiera, 2020*). Therefore,

**eLife digest** Cells in the body need to control the position of the molecules and other components inside them. To do this, they use a system of proteins that work a bit like a road network. The 'roads' are tubular structures known as microtubules, while 'vehicles' are transporters, called motor proteins, that 'walk' along the microtubules.

Microtubule networks are important in all cells, but especially in neurons, which can grow very large. These cells have tree-like branches called dendrites that receive messages from other neurons. Dendrites contain different types of microtubules with many chemical modifications. These modifications consist of specific molecules or 'groups' becoming attached to or removed from the microtubules to change their properties – for example, microtubules can be 'acetylated' or 'detyrosinated'.

Motor proteins prefer different kinds of microtubules, and so understanding transport inside cells involves creating a precise roadmap showing how many of each type of microtubule exist and where they go.

Using different super-resolution microscopy techniques, Katrukha et al. created maps of the microtubules in rat neurons. These show that acetylated microtubules form a core in the centre of the dendrites, while tyrosinated microtubules (which did not undergo detyrosination) line the cell membrane of the dendrites.

Katrukha et al. then used the maps to determine that acetylated microtubules account for 65 to 70% of all microtubules, while tyrosinated microtubules make up 20 to 30%. This means that most microtubules fall into these two categories.

The work by Katrukha et al. provides one of the first quantitative estimates of the relative amount of acetylated and tyrosinated microtubules, starting to shed light on how cells control their transport network. This could ultimately allow researchers to explore how transport changes in health and disease.

tyrosinated tubulin can be regarded as a marker for freshly polymerized microtubules. Such microtubules undergo cycles of growth and shrinkage and are therefore referred to as dynamic microtubules. Following polymerization, tubulins can also acquire new chemical groups through post-translational modifications, such as acetylation and polyglutamylation. Additionally, detyrosinated tubulin can be further proteolytically processed at the C-terminal to yield delta 2-tubulin (*Paturle-Lafanechère et al., 1991*). Such modifications often accumulate on microtubules that are long-lived and resist cold-induced or drug-induced depolymerization, which are therefore termed stable microtubules.

Despite many biochemical and physiological studies underpinning the importance of various microtubule modifications (*Sirajuddin et al., 2014*; *Janke and Magiera, 2020*; *Nekooki-Machida and Hagiwara, 2020*; *Roll-Mecak, 2019*), little is known about the relative abundance and spatial organization of different microtubule subsets within neurons. In earlier work, we revealed that stable and dynamic microtubules in dendrites are organized differently and often have opposite orientations, explaining why kinesin-3 can drive efficient anterograde transport in dendrites, unlike kinesin-1 (*Tas et al., 2017*). Nonetheless, many important aspects of the neuronal microtubule array have remained unexplored. First, do tyrosination and acetylation mark two clearly defined subsets or are there also subsets that are both highly tyrosinated and acetylated? Furthermore, what is the three-dimensional organization of different subsets and their relative abundance? Finally, do acetylation and tyrosination together mark all microtubules or are there additional subsets that carry neither of these groups? Although microtubule organization in dendrites has previously been studied using electron microscopy (*Baas et al., 1988*; *Kubota et al., 2011*), this method is difficult to combine with selective markers and therefore cannot robustly identify and map microtubule subsets throughout dendrites.

Here, we use a variety of super-resolution techniques to explore the quantitative and spatial distribution of microtubule subsets in dendrites. We find that acetylated microtubules accumulate in the core of the dendritic shaft, surrounded by a shell of tyrosinated microtubules. High-resolution microscopy enabled frequent detection of individual microtubule segments, which could be used to

carefully quantify the tyrosination and acetylation levels of these segments. This revealed that these two modifications are anti-correlated and define two distinct microtubule subsets. In addition, it enabled us to estimate the absolute number of acetylated and tyrosinated microtubules within dendrites, which revealed that they account for 65–75% and ~20–30% of all microtubules, respectively, leaving only few microtubules that do not fall in either category. Together, these results provide new quantitative insights into the uniquely organized dendritic microtubule network and help to understand the spatial regulation of neuronal transport.

## Results

We started by mapping the spatial distribution of acetylated and tyrosinated microtubules throughout the dendrite using both 2D and 3D stimulated emission depletion (STED) microscopy. Consistent with our earlier observations, this revealed that acetylated microtubules in DIV9 neurons tend to be distributed closer to the central axis of the dendrite, while the tyrosinated microtubules seem to be enriched at the outer surface, close to the membrane (*Figure 1A,B*). To quantify this observation, we built radial distribution maps of the intensity of acetylated and tyrosinated microtubules along the dendrite, which could be averaged to quantitatively describe the radial distribution of these two subsets (*Figure 1C,D*, *Video 1*). This revealed that the differential spatial organization was maintained along the length of an individual dendrite (*Figure 1D,E*) and for dendrites with various diameters (*Figure 1—figure supplement 1*), independent of STED imaging modality (*Figure 1F,G*). Whereas the intensity of both total tubulin and these two subsets decreased as the dendrite became smaller (displaying a quadratic dependence on dendrite width), the relative intensity of both subsets was nearly constant along the dendrite (*Figure 1—figure supplement 3*). Besides acetylation and tyrosination we also tried detecting detyrosinated tubulin. However, whereas the antibody that we used did reveal clear overlap between acetylation and detyrosination in non-neuronal cells, we did not obtain reliable staining in neurons, (*Figure 1—figure supplement 4*). Delta 2-tubulin was found on a subset of microtubules that overlapped with the most central part of the acetylated bundles, but was not further explored here (*Figure 1—figure supplement 4*).

We next attempted to quantify the absolute number of acetylated and tyrosinated microtubules. This cannot be achieved by just comparing fluorescent intensities, because staining efficiencies and fluorophore properties differ for each subset and need to be rescaled using single microtubules of each type as a reference. However, we were unable to distinguish individual microtubules within axons or dendrites, since (as it is known from electron microscopy studies) the distance between adjacent microtubules is often smaller than the resolution of STED (*Figure 1—figure supplement 3*; *Baas et al., 1988*; *Baas et al., 1989*). Microtubules in the cell body, however, were more dispersed and here individual microtubules could often be resolved (*Figure 2A,B*). We therefore set out to develop a workflow to enable the robust quantification of acetylation and tyrosination levels on individual microtubules, which could subsequently be used to determine the number of acetylated and tyrosinated microtubules in dendrites.

As a first step, we performed three-color STED microscopy in the soma and dendrites of DIV9 cells to detect total (alpha-)tubulin, tyrosinated tubulin, and acetylated tubulin. For the analysis of singe-microtubule intensities, we used a subvolume of the cell body just below the nucleus (*Figure 2A,B*), where the majority of microtubules were located in the *x,y* plane and confined to a relatively thin flat layer. We established a custom curvilinear structure detection algorithm to detect filament segments in all three channels and to quantify their background-corrected fluorescence intensity for all channels (*Figure 2B–D*, *Figure 2—figure supplement 1*; *Steger, 1998*).

Next, we focused on the robust estimation of average single filament intensity in the total tubulin channel. We observed that the average intensity of total tubulin was slightly lower for segments detected using acetylated tubulin, compared to segments detected using either total and tyrosinated tubulin (*Figure 2—figure supplement 1*). Possibly, the subset characterized by acetylation is highly modified and therefore stained less effectively by the alpha-tubulin antibody. To take this into account, we pooled together the total tubulin intensities of all segments, independent of the detection channel (*Figure 2E*). This histogram displayed a skewed distribution with a tail in a range of higher intensities (*Figure 2E*). We reasoned that the peak of the distribution represents the intensities of single microtubules, while the tail corresponds to the presence of microtubule bundles containing two or more overlapping microtubules, as both of these groups could clearly be

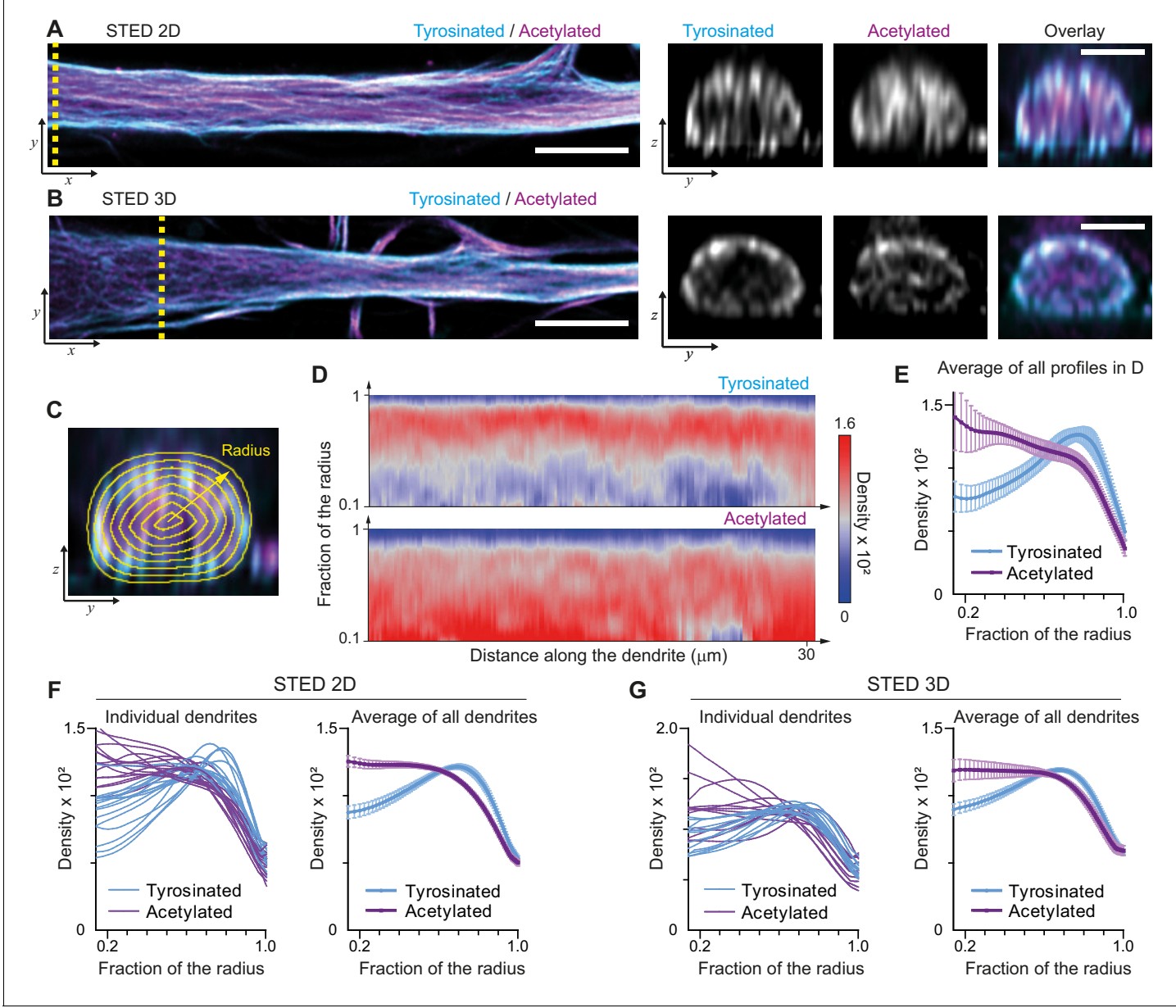

**Figure 1.** Radial distribution of microtubule subtypes in dendrites imaged using STED microscopy. (A-B) Representative single planes in XY (left) and YZ cross-sections along the yellow dashed line (right) of a dendrite imaged with 2D (A) and 3D (B) STED. Scale bar 5 μm (XY) and 2 μm (YZ). (C) Quantification of the radial intensity distribution in YZ cross-sections. The outer yellow contour marks the outline of a dendrite and concentric smaller circles represent contours of smaller radius used for quantification. (D) Heatmaps of normalized radial intensity distributions for tyrosinated (top) and acetylated (bottom) microtubules along the length of the dendrite shown in (A). (E) Radial distribution of tyrosinated (cyan) and acetylated (magenta) microtubule posttranslational modifications averaged over the length of the dendrite shown in (A) (n=176 profiles). Error bars represent SD. (F-G) Radial distribution of modifications averaged per dendrite (left) and over all dendrites (right) imaged using 2D STED (panel (F), n=4971 profiles, 15 cells, N=two independent experiments) or 3D STED (panel (G), n=5891 profiles, 12 cells, N=two independent experiments). Error bars represent S.E.M. The online version of this article includes the following figure supplement(s) for figure 1:

**Figure supplement 1.** Thickness of dendrites imaged using STED.

**Figure supplement 2.** Resolution of STED.

**Figure supplement 3.** Intensity analysis along the whole length of individual dendrites.

**Figure supplement 4.** Detyrosinated and delta-2 microtubules subsets.

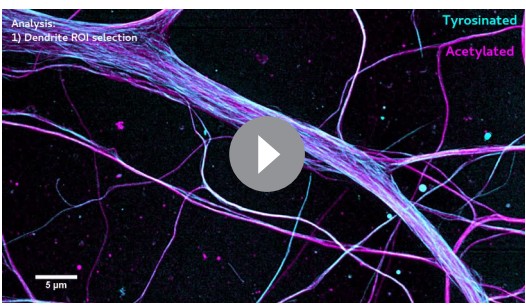

**Video 1.** Illustration of radial intensity distribution analysis in dendrites acquired using STED microscopy. https://elifesciences.org/articles/67925#video1

distinguished in the original images (*Figure 2B, D*). To obtain a robust estimate for the intensity of individual microtubules, we fitted the histograms with a sum of two Gaussian distributions (*Figure 2E*). The first Gaussian represents the intensity distribution of a single microtubule, with a standard deviation determined by several different factors (antibody staining variations, microtubules going in and out of focus). The second Gaussian corresponds to bundles of two microtubules and mathematically represents a convolution of first Gaussian with itself (assuming random independent intensity sampling of two microtubules in a bundle). We used the mean value of the first Gaussian as an estimate of average single microtubule intensity in the total channel.

We proceeded with an estimation of the average levels of tyrosination and acetylation of an individual microtubule (per cell). To exclude segments corresponding to the bundles of multiple microtubules, we only analyzed segments for which the total tubulin intensity was below the mean intensity of the first Gaussian plus one standard deviation (*Figure 2E*). Visual inspection of segments below and above the threshold confirmed that this filtering eliminated the majority of thick or bright bundles (*Figure 2F*). Consistently, including only the single-MT segments determined from total tubulin intensities also resulted in more unimodal and symmetric distributions for the intensities of tyrosination and acetylation levels of segments identified in these respective channels (*Figure 2G*, *Figure 2—figure supplement 1*). The average intensities of single tyrosinated or acetylated microtubule were estimated as the average values of intensities detected and quantified in the same corresponding channel. These values were used for the normalization of intensities shown at *Figure 2G* and *Figure 2—figure supplement 2*. In addition, we also quantified the levels of tyrosination and acetylation of all segments detected in the acetylation and tyrosination channel, respectively. This analysis enabled us to build two two-dimensional histograms that show the levels of both tyrosination and acetylation for microtubule segments detected either in the acetylation channel or the tyrosination channel (*Figure 2G*).

The resulting histograms show that segments detected by acetylation have, on average, lower levels of tyrosination than segments detected by tyrosination, and vice versa (*Figure 2G*). This quantitatively confirms the general impression that these chemical groups mark two different subsets and that microtubules with high levels of acetylation are mostly detyrosinated. However, despite clearly separating into two subsets, even highly acetylated microtubules display residual tyrosination, whereas many tyrosinated microtubules have some extent of acetylation. The measured relative level of tyrosination for acetylated microtubules, compared to average tyrosination of tyrosinated microtubules, which we termed $\alpha$, was equal to $0.53 \pm 0.11$ (average $\pm$ SD) and the level of acetylation for tyrosinated microtubules, compared to the acetylation of acetylated microtubules, termed $\beta$, was $0.45 \pm 0.07$ (*Figure 2H*). These results demonstrate that microtubules can be divided in two different subsets based on the detection of tyrosination and acetylation. We will refer to microtubules detected in the acetylated tubulin channel, displaying on average 47% lower levels of tyrosination than microtubules detected in the tyrosinated channel, as stable microtubules. Likewise, dynamic microtubules are identified as microtubules detected in the tyrosinated tubulin channel and feature 55% lower levels of acetylation than microtubules detected in the acetylated channel.

We next set out to use the intensities of total tubulin, acetylation, and tyrosination on individual microtubules to determine the both the total number of microtubules within dendrites, as well as the number of stable and dynamic microtubules within dendrites. To estimate the total number of microtubules, the dendritic intensity of total tubulin was divided by the single-microtubule intensity (assuming consistent labeling throughout the neuron). However, for the quantification of stable and dynamic microtubules, we needed to correct for the 'chemical crosstalk' that we observed, that is the tyrosination and acetylation levels detected for stable and dynamic microtubules, respectively. As a result, the integrated tyrosinated intensity of a dendrite was not just the sum of intensities of dynamic microtubules, but also included the contribution from the residual tyrosination of stable

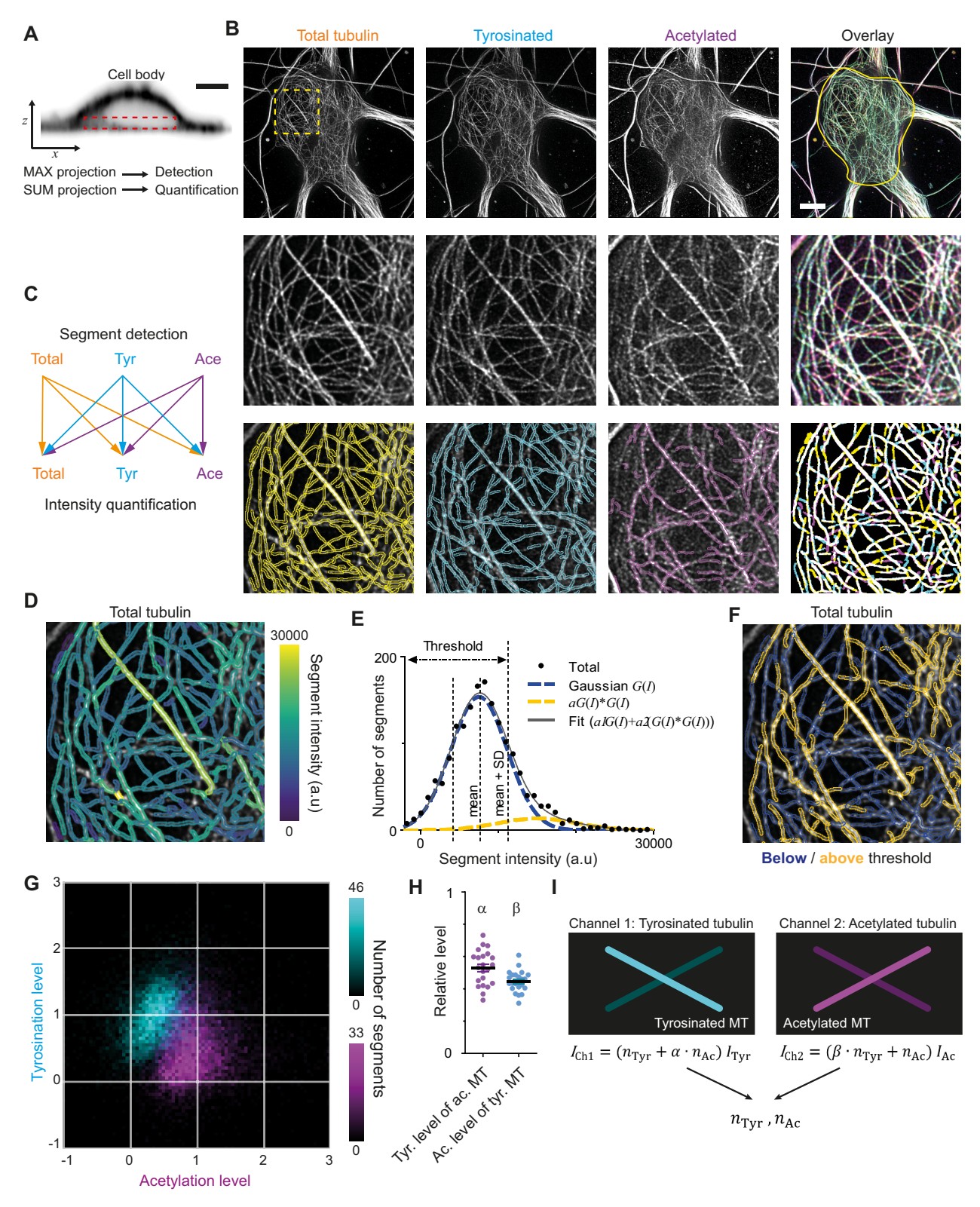

**Figure 2.** Analysis pipeline for detection and intensity quantification of individual (post-translationally modified) microtubules. (**A**) Vertical cross-section along a neuronal cell body (soma). Dashed rectangle marks the volume (sub-z-stack) under the nucleus used for microtubule filament detection (maximum intensity projection) and quantification (sum of all slices). Scale bar 5 μm. (**B**) Top row: Maximum intensity projection of 2D STED z-stacks of DIV9 neurons stained for alpha-tubulin (total) and for tyrosinated tubulin and acetylated tubulin. Solid yellow contour in the overlay marks the area used

*Figure 2 continued on next page*

*Figure 2 continued*

for detection of individual filaments. Scale bar 5 µm. Middle row: Zoom-ins corresponding to the dashed yellow square in the top row. Bottom row: Example of individual filament detection in each channel (first three panels) and binarized overlay of the detections (right panel). (C) Schematics of individual filament analysis: detection was performed in each channel separately using maximum intensity projection. For each detected segment, the intensity was quantified in all three channels. (D) Outlines of microtubule filaments detected and quantified in the total tubulin channel (for the cell depicted in (A)), color-coded according to their background corrected integrated intensity. (E) Histogram of the background-corrected integrated intensity of individual filaments detected in all three channels and quantified in total tubulin channel for the cell shown in (A) (black dots, n=1736). The solid black line shows the fit of the sum of two Gaussian functions: the first corresponds to a single filament intensity distribution (dashed blue line) and the second Gaussian corresponds to the double filament intensity distribution, that is the first Gaussian convoluted with itself (dashed orange line). Dashed lines mark the mean and mean plus standard deviation of the first Gaussian. The latter was used as a threshold for single microtubule filtering. (F) Illustration of single filament intensity filtering: outlines of the filaments with intensity below the threshold are colored in blue (assigned as a single microtubule) and filaments above it in orange (assigned as two or more microtubule bundles). (G) Two-color heatmap overlay of normalized intensity distributions of single microtubule segments detected in tyrosinated (cyan, n=10281 segments) and acetylated (magenta, n=9369 segments) channels and quantified in both (22 cells, N=2 independent experiments). (H) Average normalized level of tyrosination per cell for single microtubule segments detected in the acetylated channel ($\alpha$, magenta) and average normalized level of acetylation for segments detected in tyrosinated channel ($\beta$, cyan). Horizontal black lines mark mean ± S.E.M. (22 cells, N=2 independent experiments). (I) Illustration of the analysis pipeline for the quantification of tyrosinated and acetylated microtubules number in dendrites.

The online version of this article includes the following figure supplement(s) for figure 2:

**Figure supplement 1.** Analysis of microtubule segments intensities.

**Figure supplement 2.** Heatmaps of microtubule subsets using STED.

microtubules (and vice versa). This situation was analogous to instances of spectral crosstalk in fluorescence microscopy, where emission from one dye is detected in the spectral channel of another dye (*Zimmermann, 2005*), and we therefore used standard formulas for spectral unmixing and our estimates for $\alpha$ and $\beta$ (*Figure 2H,I*) to take this posttranslational modification crosstalk into account.

When we calculated the composition of the dendritic microtubule network, we focused on the proximal 5–10 µm of a dendrite (*Figure 3A,B*) and used the corresponding single-filament intensity and crosstalk estimates from the soma of the same cell. First of all, this showed that the total number of microtubules in a dendrite depends linearly on its cross-sectional area in the range from 1 to 10 $\mu m^2$ with a slope of 68 microtubules per $\mu m^2$. In addition, it revealed that dendrites have over four times more acetylated microtubules than tyrosinated microtubules (74 ± 8% versus 16 ± 11%, average ± SD) and that this factor was largely independent of the diameter of the dendrite (*Figure 3C, D,E*). We furthermore found that these two subsets did not completely account for the total number of microtubules that we measured, leaving a small fraction of 10 ± 14% of microtubules that were classified as neither acetylated nor tyrosinated.

A potential weakness of the analysis that we performed is that it assumes that the measured levels of acetylation and tyrosination on stable and dynamic subsets found in the cell body are comparable to those found within dendrites. To overcome this, single-microtubule levels of acetylation and tyrosination would need to be measured directly in the dendrites, which requires 3D images in which single dendritic microtubules are clearly distinguishable. Because this was not possible using our STED microscopy approach, we switched to expansion microscopy (ExM) to improve both lateral and axial resolution (*Jurriens et al., 2021*; *Tillberg et al., 2016*). In expansion microscopy, stained samples are embedded in and crosslinked to a swellable hydrogel, followed by proteolytic digestion and physical expansion, which will increase the spacing between the remaining gel-linked protein fractions and fluorophores. Since gels expand in all dimensions, this leads to an isotropic resolution improvement determined by the expansion factor of the gel (about four times).

Indeed, expanded samples demonstrated a substantial increase in the clarity with which microtubule organization could be perceived (*Figure 4A,B*, *Figure 4—figure supplement 1*, *Video 2*). We therefore repeated our analysis of the spatial distribution of tyrosinated and acetylated microtubules and found that the peripheral enrichment of tyrosinated microtubules was even more pronounced in ExM samples, as shown in *y,z* cross-section images (*Figure 4B*) and radial distribution plots (*Figure 4C,D,E,F*). Even though visual tracing of individual filaments remained challenging (*Figure 4—figure supplement 1*), we were able to estimate the relative abundance of acetylated and tyrosinated microtubules by decomposing the radial density of total tubulin as a sum of the acetylated and tyrosinated radial densities (*Figure 4G*). Although this analysis does not take into account

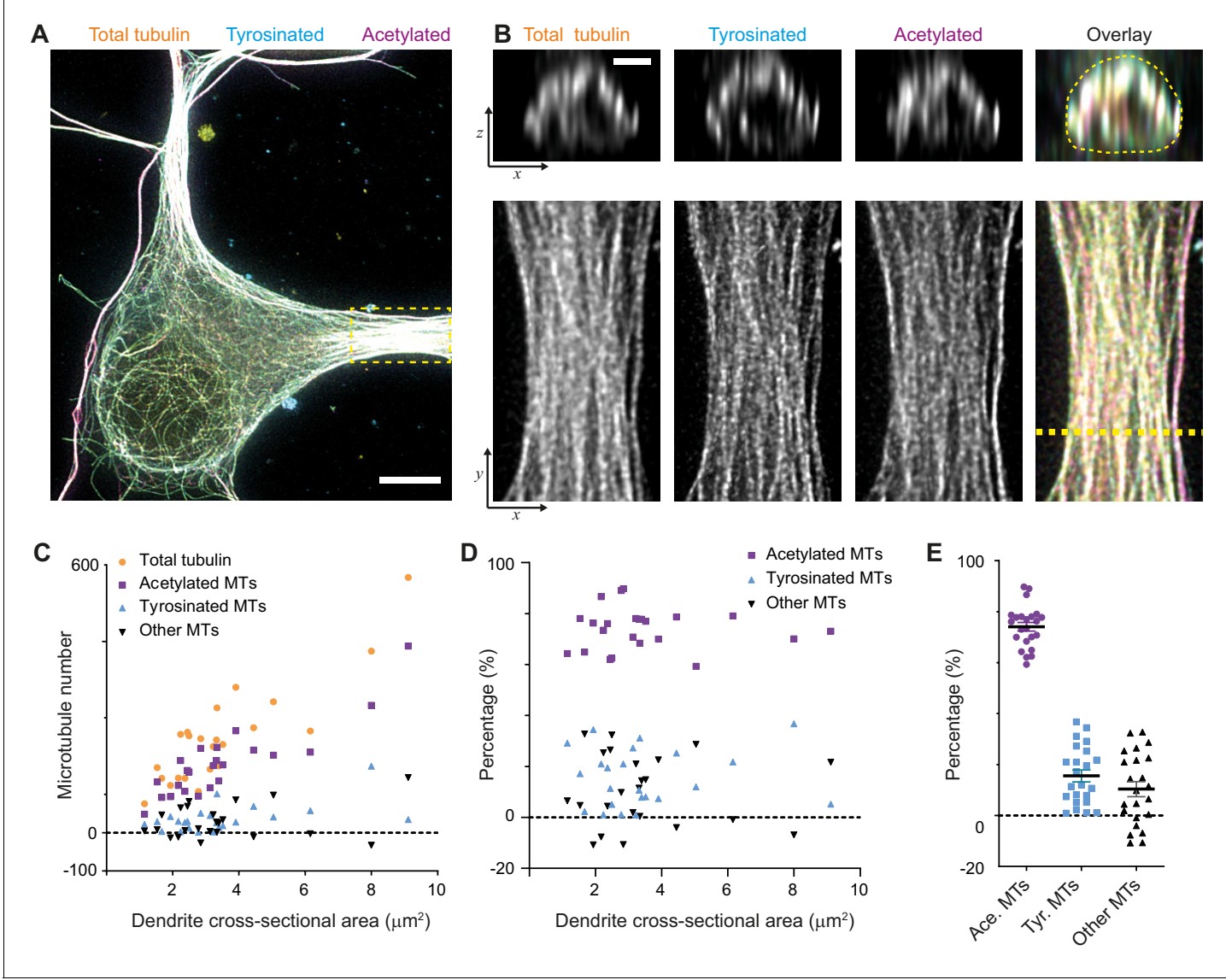

**Figure 3.** Estimation of absolute numbers of (modified) microtubules in dendrites using single-microtubule intensities from the soma. (A) Three-color overlay of maximum intensity projection of a 2D STED z-stack including the whole volume of dendrites (total tubulin in yellow, tyrosinated in cyan and acetylated in magenta). Dashed yellow rectangle marks zoom-in shown in (B). Scale bar 5 μm. (B) Maximum intensity projections of a segment of dendrite (bottom row, marked by yellow dashed rectangle in (A)) and individual YZ slices (top row, corresponds to a dashed yellow line). (C) Numbers of total, tyrosinated, acetylated and other (i.e. neither tyrosinated nor acetylated) microtubules per dendrite as a function of cross-sectional area (n=23 dendrites, N=2 independent experiments). These numbers were determined using the approach outlined in *Figure 2I*. (D-E) Percentage of tyrosinated, acetylated and other microtubules per dendrite as a function of dendrite's cross-section area (D) or pooled together (E). Horizontal black lines in (E) mark mean ± S.E.M. (n=23 dendrites, N=2 independent experiments).

the fraction of microtubules that is neither tyrosinated or acetylated, it independently confirms the prevalence of acetylated microtubules (65%) over tyrosinated (35%).

The successful decomposition of total tubulin using only these two subsets, as well as the higher fraction of tyrosinated tubulin in comparison with our estimate from soma-based intensity rescaling (*Figure 3E*), prompted us repeat our microtubule counting using dendrite-based intensity rescaling. Unfortunately, our ExM data also did not have sufficient resolution to resolve enough individual microtubules to reliably determine the single-microtubule estimates of acetylation and tyrosination required for such analysis. First, we tried to use ExSTED microscopy to improve resolution

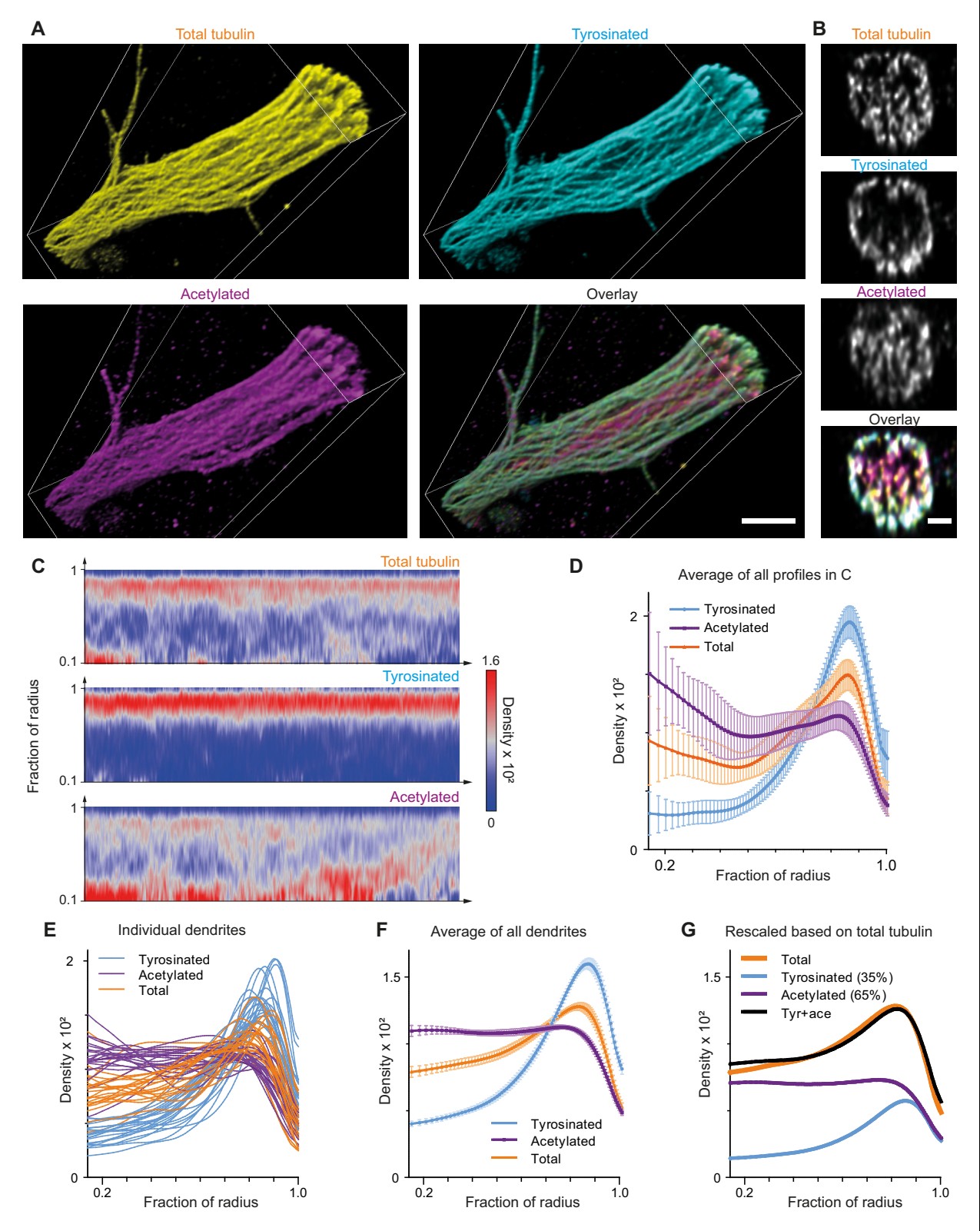

**Figure 4.** Expansion Microscopy improves quantification of the radial distribution of microtubule modifications in dendrites. (A) Representative volumetric 3D rendering of total tubulin and its posttranslational modifications in a dendrite imaged using ExM. Scale bar 2 μm (physical size post-expansion 8.3 μm). (B) Representative single YZ cross-section of the dendrite from (A). Scale bar 0.5 μm (physical size post-expansion 2.07 μm). (C) Heatmaps of normalized radial intensity distributions for total tubulin (top), tyrosinated (middle) and acetylated (bottom) microtubule

*Figure 4 continued on next page*

*Figure 4 continued*

posttranslational modifications along the length of the dendrite shown in (**A**). Abscissa units are recalculated according to expansion factor (17 μm equals to 70.5 μm physical size post-expansion). (**D**) Radial distribution of total tubulin (orange), tyrosinated (cyan) and acetylated (magenta) microtubule posttranslational modifications averaged over the length of the dendrite shown in (**A**) (n=404 profiles). Error bars represent SD. (**E-F**) Radial distribution of total tubulin (orange) and tyrosinated (cyan) and acetylated (magenta) posttranslational modifications intensities averaged per dendrite (**E**) and among all dendrites (**F**) imaged using ExM (n=9460 profiles, 22 cells, N=2 independent experiments). Error bars represent S.E.M. (**G**) Decomposition of total tubulin radial intensity distribution as a weighted sum of tyrosinated and acetylated distributions.

The online version of this article includes the following figure supplement(s) for figure 4:

**Figure supplement 1.** Characterization of dendrites imaged using ExM.

(*Gao et al., 2018*) and found that three-color volumetric STED acquisition of ExM samples resulted in substantial photobleaching, which strongly impaired the integrated intensity analysis (data not shown). We then realized that most dendritic microtubules are organized in bundles that run parallel to the coverslip and thus resolving microtubules would be easier if we could alter the sample orientation such that microtubules are aligned with the optical axis of our microscope. Since in a regular ExM acquisition the axial dimension (the poorest) of the PSF is oriented perpendicular to the filaments (located parallel to the coverslip plane), we decided to generate thick gel slices that were rotated by 90 degrees, a procedure we termed FlipExM (*Figure 5A,B*). In this configuration, we exploit the better lateral resolution to resolve individual microtubules, while PSF blurring along the optical axis happens parallel to filaments (*Figure 5B*, *Videos 3* and *4*).

Although we still could not discern individual microtubules within tight bundles, we observed many individual microtubules traversing through the dendritic volume. We therefore set out to quantify the intensities of these microtubules, so that these could be used to quantify the total number of microtubules and the abundance of microtubule subsets within the dendrite. The cross-sections of individual microtubule filaments were automatically detected in each channel in dendrites cross-sections (*Figure 5B*, bottom row), and we quantified their area and their background-corrected intensity in each channel. To exclude noise and bundles, we then applied area and roundness filters on our detections (*Figure 5C,D*). After this geometrical filtering, the intensity distribution showed a similar bimodal or skewed shape as found earlier for the filaments in the cell body (*Figure 5E*, *Figure 2E*, *Figure 5—figure supplement 1*). Therefore, we again used curve fitting (similar to *Figure 2*) to estimate the average intensity of microtubule cross-sections in each channel. The distributions of tyrosinated and acetylated detections in the tyrosination/acetylation plane (*Figure 5F*, *Figure 5—figure supplement 2*) appeared very similar to the data obtained earlier using the cell body (*Figure 2G*), but with more distinct separation between the two clusters. Compared to the cell body data, we found slightly different values for the average tyrosination level of acetylated microtubules (0.45 ± 0.05), as well as the acetylation level for tyrosinated microtubules (0.60 ± 0.17) (*Figure 5G*).

Finally, we used the single-microtubule intensity levels measured directly within dendrites to quantify total microtubule numbers, as well as the number of acetylated and tyrosinated microtubules (*Figure 5G–J*). First, we divided the integrated cross-section intensity of the total tubulin channel by our single cross-section intensity estimate for dendrites with different diameters (*Figure 5G–J*). A linear fit through the total number of microtubules as a function of cross-sectional area yields an estimated microtubule density of 68 and 53 microtubules per square micrometer for the cell body and dendrite methods, respectively (*Figure 5J*). Next, to determine the number of acetylated and tyrosinated microtubules, we employed the 'modification unmixing' approach mentioned previously. Consistent with our earlier results, this analysis revealed that stable, acetylated microtubules form the largest

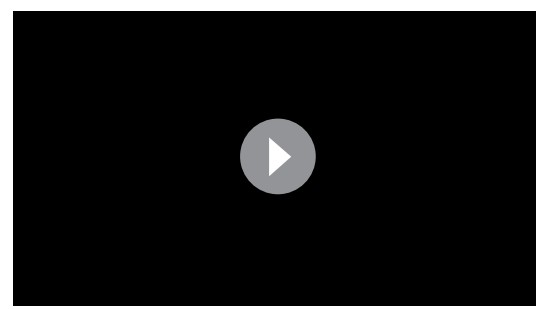

**Video 2.** 3D volumetric rendering of a dendrite imaged using ExM (same as in *Figure 4A*). Scale bar corresponds to the physical post expansion size. https://elifesciences.org/articles/67925#video2

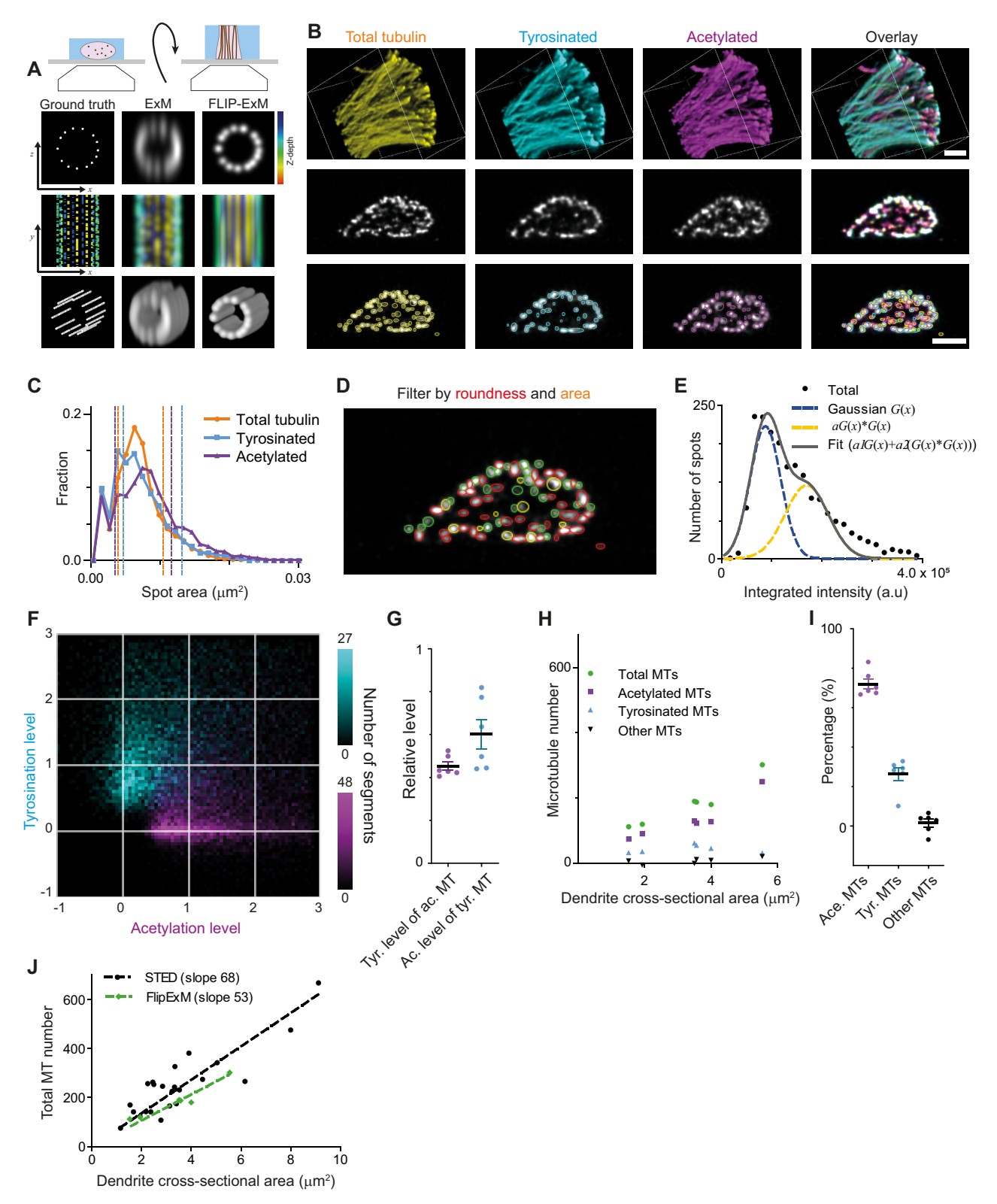

**Figure 5.** Direct estimation of microtubule numbers in dendrites using FlipExM. (**A**) Top: Schematics of gel reorientation for FlipExM imaging. Bottom: Simulated z-stacks illustrating the advantages of FlipExM for imaging of dendritic microtubules. A set of simulated circumferential microtubules in a dendrite of 1 µm diameter (left) were convoluted with a point spread function corresponding to a regular ExM (middle) or FlipExM (right) imaging (top to bottom: single XZ plane, color-coded depth projection, 3D rendering). (**B**) Representative volumetric 3D rendering (top) and single YZ slices (middle)

*Figure 5 continued on next page*

*Figure 5 continued*

of total tubulin and its posttranslational modifications in a dendrite imaged using FlipExM. The bottom row shows automatic detections of microtubules in cross-sections. Scale bars 1 µm (physical size post-expansion 4.15 µm). (C) Area histogram of spots corresponding to microtubules cross-sections in three channels for the dendrite shown in (B). Spots were pre-filtered using roundness criteria (n=7103, 4844, 3821 for total, tyrosinated and acetylated channels). An interval between dashed lines marks the range for spot's area filtering. Abscissa units are recalculated according to expansion factor (4.15). (D) Single YZ slice of total tubulin channel with detections filtered by roundness marked by red circles, detections filtered by area marked by yellow circles and remaining detections attributed to single microtubules marked in green. (E) Histogram of background subtracted integrated intensity of individual microtubules cross-sections detected and quantified in total tubulin channel for the dendrite shown in (B) (black dots, n=1981). The solid black line shows the fit of sum of two Gaussian functions: the first corresponds to a single microtubule cross-section intensity distribution (dashed green line) and the second Gaussian corresponds to the double cross-section intensity distribution, that is first Gaussian convoluted with itself (dashed red line). (F) Two color heatmap overlay of normalized intensity distributions of single microtubule cross-sections detected in tyrosinated (cyan, n=8642 spots) and acetylated (magenta, n=12552 spots) channels and quantified in both (6 cells, N=2 independent experiments). (G) Average normalized level of tyrosination per cell for single microtubule cross-section detected in acetylated channel (α, magenta) and average normalized level of acetylation for cross-sections detected in tyrosinated channel (β, cyan). Horizontal black lines mark mean ± S.E.M. (6 cells, N=2 independent experiments). (H) Numbers of total, tyrosinated, acetylated, and non-modified microtubules per dendrite depending on dendrite's cross-section area (n=6 cells, N=2 independent experiments). (I) Percentage of tyrosinated, acetylated, and non-modified microtubules as a fraction of total microtubule number per dendrite. Horizontal black lines mark mean ± S.E.M. (n=6 cells, N=2 independent experiments). (J) Number of microtubules per dendrite determined using STED (black dots, data from *Figure 2C*) or FlipExM (green dots, data from (H)) imaging. Dashed lines show independent linear fits passing through the origin.

The online version of this article includes the following figure supplement(s) for figure 5:

**Figure supplement 1.** Estimation of single microtubule intensity per channel.

**Figure supplement 2.** Heatmaps of microtubule subsets using FlipExM.

**Figure supplement 3.** Percentage of microtubule subsets per dendrite.

**Figure supplement 4.** Number of microtubules along the length of a dendrite.

**Figure supplement 5.** Different crosstalk estimation methods for FlipExM data.

population (72 ± 6%). The fraction of tyrosinated microtubules was larger than our earlier estimate (26 ± 8%), at the expense of the fraction of microtubules that were neither acetylated nor tyrosinated (2 ± 5%, *Figure 5H,I*, *Figure 5—figure supplement 3*). These results indicate that acetylated and tyrosinated microtubules together account for 98% of all dendritic microtubules, with acetylated microtubules being almost three times more abundant.

## Discussion

The high density of the neuronal microtubule cytoskeleton has so far obscured its exact composition and organization. Earlier work has used electron microscopy to reveal the number and spatial organization in dendrite cross-sections, but this technology is difficult to combine with the robust detection of distinct subsets (*Baas et al., 1988*; *Kubota et al., 2011*). While early work on axonal microtubules successfully detected modified microtubules using immunoelectron microscopy, these methods have not been systematically applied to dendrites (*Baas and Black, 1990*; *Ahmad et al., 1993*). Here, we used super-resolution light microscopy to explore the composition and architecture of the microtubule cytoskeleton in dendrites. In addition to visualizing all microtubules using an antibody against alpha-tubulin, we focused on two microtubule subsets: those labeled using antibodies against acetylation and tyrosination, typically classified as stable and dynamic microtubules (*Janke and Magiera, 2020*; *Guardia et al., 2016*; *Schulze and Kirschner, 1987*). Volumetric STED and expansion microscopy revealed a striking spatial organization in which stable, acetylated microtubules are enriched in the core of the dendritic shaft, surrounded by a shell of dynamic, tyrosinated microtubules (*Figures 1* and *4*). While our earlier two-dimensional super-resolution imaging

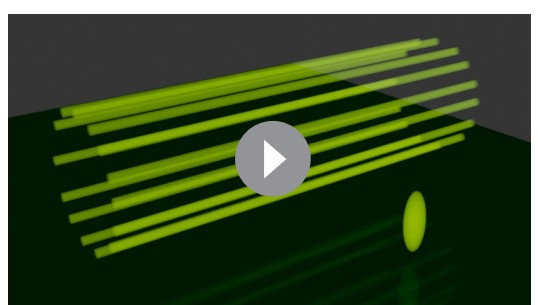

**Video 3.** Illustration of gel sample reorientation and microscope's PSF for FlipExM imaging.
https://elifesciences.org/articles/67925#video3

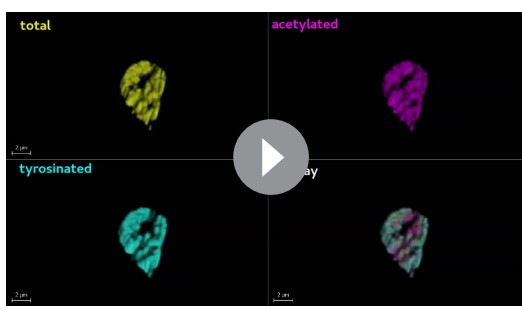

**Video 4.** 3D volumetric rendering of a dendrite imaged using FlipExM (same as in *Figure 5B*). Scale bar corresponds to the physical post expansion size. https://elifesciences.org/articles/67925#video4

already hinted at a spatial separation between different subsets (*Tas et al., 2017*), the current work provides the first quantitative three-dimensional mapping of subset organization throughout a large set of dendrites.

The enrichment of dynamic microtubules near the plasma membrane is consistent with the well-established interplay between growing microtubule plus ends and (sub)cortical complexes (*van de Willige et al., 2016*; *Akhmanova and Steinmetz, 2015*). More specifically, dynamic microtubules have been shown to regularly invade into dendritic spines to facilitate intracellular transport or regulate spine morphology in response of specific synaptic stimuli (*Esteves da Silva et al., 2015*; *Jaworski et al., 2009*; *McVicker et al., 2016*; *Hu et al., 2008*; *Schätzle et al., 2018*). Next to ensuring the enrichment of dynamic microtubules near the plasma membrane, spatial separation between stable and dynamic microtubules might also promote efficient intracellular transport by separating cargoes driven by subset-specific motors. Moreover, for motors that do not discriminate between microtubule subsets, the spatial separation between mostly minus-end out oriented stable microtubules and mostly plus-end out oriented dynamic microtubules (*Tas et al., 2017*) will facilitate directional transport by limiting directional switching induced by cargo-attached motors binding to neighboring microtubules of opposite polarity. In future work, we will explore how the transport patterns of different cargoes depend on the associated motors and the organization of the neuronal microtubule cytoskeleton.

The use of three-color super-resolution imaging allowed us to include a marker for total tubulin and, in combination with novel analysis methods, provide two independent estimates for the total number of microtubules in dendrite sections, as well as the number of acetylated and tyrosinated microtubules (*Figure 3*, *Figure 5*). Our estimates for the total microtubules were obtained by dividing the total intensity of a generic tubulin stain by the intensity measured on individual microtubules in either the soma (STED, *Figure 3*) or dendrite itself (Flip-ExM, *Figure 5*), which revealed an average density of 68 or 53 microtubules per $\mu m^2$, respectively. These values are consistent with earlier estimates using electron microscopy (66 microtubules per $\mu m^2$)(*Kubota et al., 2011*). Although we used various filtering steps to prevent mistaking small microtubule bundles for individual microtubules, it remains possible that occasional inclusion of such bundles increased our estimate for single microtubules, thereby lowering our estimate for the total number of microtubules. Alternatively, these differences could be caused by local differences in expansion factor or just reflect sample-to-sample differences in the number of microtubules per area. Nonetheless, the close correspondence between our estimates and values obtained using electron microscopy on dendritic cross-sections demonstrates the strength of combining super-resolution microscopy with quantitative analysis.

In all our estimates, acetylated microtubules strongly outnumber tyrosinated microtubules. The two single-microtubule calibration methods (i.e. soma versus dendrite) yielded strikingly similar estimates for the percentage of acetylated microtubules (74 ± 8% versus 72 ± 6%), while their estimates for tyrosinated and other (non-tyrosinated and non-acetylated) microtubules differed to some extent (16% tyrosinated and 10% other microtubules for soma versus 26% and 2% for dendrite estimations, respectively). These results suggest that the acetylation level of stable microtubules is similar between soma and dendrites, whereas the tyrosination level of dynamic microtubules could be higher in the soma than the dendrite. For the soma-based method, this would overestimate the dendritic single-microtubule tyrosination levels and result in undercounting dendritic tyrosinated microtubules, leaving a larger fraction of *other* microtubules. The idea that dendritic dynamic microtubules are on average more detyrosinated is consistent with our finding that these microtubules also have higher levels of acetylation compared to the soma (*Figure 5G*). We therefore consider the dendrite-intrinsic measurements to be more reliable, that is 72 ± 6% acetylated, 26 ± 8% tyrosinated and 2 ± 5% other microtubules.

It is important to note that the two markers that we used, acetylation and tyrosination, represent only a small part of the possibly ways in which the microtubule surface can become differentiated, such as through other modifications like polyglutamylation, phosphorylation, palmitoylation, incorporation of different tubulin isoforms, and adsorption of different MAPs (*Sirajuddin et al., 2014*; *Janke and Magiera, 2020*). We focused on acetylation and tyrosination because these modifications label clearly distinct subsets and display the most direct correlation with microtubule stability (e.g. only acetylated microtubules remain after nocodazole treatment (*Tas et al., 2017*) and motor selectivity (e.g. Kinesin-1 binding correlates with acetylation (*Tas et al., 2017*; *Jansen et al., 2021*)). Other modifications, such as glutamylation, are more rheostatic and are known to play different roles at different levels (*Roll-Mecak, 2019*; *Valenstein and Roll-Mecak, 2016*). Importantly, the binary classification scheme used to classify microtubules as either acetylated or tyrosinated is most likely an oversimplification that does not do full justice to the rich modification landscape of microtubules, where also different parts of a microtubule can display different modifications (*Baas and Black, 1990*; *Ahmad et al., 1993*). Our choice for acetylation and tyrosinations was furthermore prompted by the availability of reliable antibodies, which remains a challenge for many other modifications. Remarkably, our analyses revealed that labeling acetylated and tyrosinated microtubules leaves only a very small fraction (2%) of microtubules unlabeled. This suggests that most detyrosinated microtubules in dendrites are also acetylated and that other modifications or MAPs are found on microtubules that are either tyrosinated or acetylated.

In this work, we have introduced innovative imaging and analysis approaches to quantitatively map the neuronal cytoskeleton. In future work, we aim to map how other modifications and various microtubule-associated proteins are distributed over these two subsets of microtubules. In addition, the distribution of modifications and microtubule-associated proteins along the length of individual microtubules should be mapped to better understand how dynamic microtubules may become stabilized. We anticipate that such experiments will benefit from ongoing advances in expansion microscopy, such as iterative expansion or single-step approaches with higher expansion factors.

# Materials and methods

## Key resources table

| Reagent type (species) or resource | Designation | Source or reference | Identifiers | Additional information |
|---|---|---|---|---|
| Cell line (*Cercopithecus aethiops*) | COS7 | Lukas Kapitein lab | | Cell line has not been authenticated, but mycoplasm testing was done fewer than 26 weeks before experiments |
| Biological sample (*Rattus norvegicus domestica*) | Primary hippocampal neuron culture | Lukas Kapitein lab | | Cultured on embryonic day 18 |
| Antibody | Mouse monoclonal anti-Acetylated tubulin | Sigma | Cat#T7451, [6-11B-1], RRID:AB_609894 | IF Expansion (1:200), IF confocal/STED (1:600) |
| Antibody | Rat monoclonal anti-Tyrosinated tubulin | Abcam | Cat#Ab6160, [YL1/2], RRID:AB_305328 | IF Expansion (1:100), IF confocal/STED (1:250) |
| Antibody | Rabbit recombinant anti-alpha tubulin | Abcam | Cat# 52866, [EP1332Y] RRID:AB_869989 | IF Expansion (1:250), IF confocal/STED (1:500) |
| Antibody | Rabbit polyclonal anti-detyrosinated tubulin | Merck | Cat# AB3210, | IF STED (1:500) |
| Antibody | Rabbit anti-Δ2 tubulin | Millipore | Cat# AB3203, RRID:AB_177351 | IF STED (1:500) |
| Antibody | Alexa 594 Goat Anti-Rat IgG (H+L) | Molecular Probes, Life Technologies | Cat#A11007, RRID:AB_10561522 | IF Expansion (1:250), IF confocal/STED (1:500) |

*Continued on next page*

*Continued*

| Reagent type (species) or resource | Designation | Source or reference | Identifiers | Additional information |
|---|---|---|---|---|
| Antibody | Alexa 488 Goat Anti-Rabbit IgG (H+L) | Molecular Probes, Life Technologies | Cat#A11034, RRID:AB_2576217 | IF Expansion (1:250), IF confocal/STED (1:500) |
| Antibody | Alexa 594 Goat Anti-Mouse IgG (H+L) | Molecular Probes, Life Technologies | Cat#A11032, RRID:AB_2534091 | IF Expansion (1:250), IF confocal/STED (1:500) |
| Antibody | Alexa 488 Goat Anti-Rat IgG (H+L) | Molecular Probes, Life Technologies | Cat#A11006, RRID:AB_2534074 | IF Expansion (1:250), IF confocal/STED (1:500) |
| Antibody | Abberior Star 635P goat anti-mouse IgG (H+L) | Abberior GmbH | Cat#ST635P -1001–500 UG | IF Expansion (1:250), IF confocal/STED (1:500) |
| Antibody | Abberior Star 635P goat anti-rabbit IgG (H+L) | Abberior GmbH | Cat#ST635P -1002–500 UG | IF Expansion (1:250), IF confocal/STED (1:500) |
| Chemical compound, drug | Triton X-100 | Sigma | Cat#X100 | |
| Chemical compound, drug | Gluteraldehyde (8%) | Sigma | Cat#G7526 | |
| Chemical compound, drug | Paraformaledehyde (16%) | Electron Microscopy Sciences | Cat#15710 | EM-grade |
| Chemical compound, drug | Acryloyl-X | Thermo Fisher | Cat#A20770 | |
| Chemical compound, drug | Sodium Acrylate | Sigma Aldrich | Cat#408220 | |
| Peptide, recombinant protein | Proteinase K | ThermoFisher | Cat#EO0492 | |
| Software, algorithm | Huygens Professional software version 17.04 | Scientific Volume Imaging, the Netherlands | RRID:SCR_014237 | |
| Software, algorithm, | Correlescence plugin v.0.0.4 (2021) | Eugene Katrukha | | https://github.com/ ekatrukha/Correlescence https://doi.org/10.5281/zenodo.4534715 |
| Software, algorithm, | CurveTrace plugin ver.0.3.5 (2021) | Eugene Katrukha | | https://github.com/ ekatrukha/CurveTrace https://doi.org/10.5281/zenodo.4534721 |
| Software, algorithm | ComDet plugin v.0.5.3 (2021) | Eugene Katrukha | | https://github.com/ ekatrukha/ComDet https://doi.org/10.5281/zenodo.4281064 |
| Other | Silicone mold for gels 13 mm | Sigma-Aldrich | Cat#GBL664107 | |

## Primary rat neuron culture and immunostaining

Dissociated hippocampal neuron cultures were prepared from embryonic day 18 rat pups of mixed gender according to the previously published protocol (*Kapitein et al., 2010*). Briefly, cells were plated on 18-mm glass coverslips coated with poly-l-lysine (37.5 mg/ml) and laminin (1.25 mg/ml) in a 12-well plate at a density of 50 k/well. Cultures were maintained in Neurobasal medium (NB) supplemented with 2% B27, 0.5 mM glutamine, 16.6 µM glutamate, and 1% penicillin/streptomycin at 37°C in 5% $CO_2$. COS7 cells were cultured in DMEM supplemented with 10% fetal bovine serum and 1% penicillin/streptomycin.

We followed our recently published immunostaining/expansion protocol, described in details in *Jurriens et al., 2021*. In short, at DIV9 neurons were pre-extracted for 1 min using 500 µl of 0.3% Triton X-100 (Sigma X100), 0.1% glutaraldehyde (Sigma G7526) in MRB80 buffer (80 mM Pipes

(Sigma P1851), 1 mM EGTA (Sigma E4378), 4 mM MgCl$_2$, pH 6.8) and fixed for 10 min using 4% PFA (Electron Microscopy Sciences 15,710) and 4% sucrose in MRB80 buffer (both solutions were pre-warmed to 37˚C). Fixed neurons were washed three times in PBS and permeabilized for 10 min in 500 µl of 0.25% Triton X-100 in MRB80 buffer. Samples were further incubated in 500 µl of blocking buffer (3% w/v BSA in MRB80 buffer) for at least 45 min at room temperature. Finally, fixed neurons were sequentially incubated for two hours at room temperature (or overnight at 4˚C) with primary and secondary antibodies diluted in blocking buffer (3% w/v BSA in MRB80 buffer) and washed three times in PBS. The same fixation protocol was used for staining with COS7 cells. We used the following combinations of primary (dilution 1;500 for STED and confocal; dilution 1:200 for expansion) and secondary (dilution 1;500 for STED and confocal; dilution 1:250) antibodies: mouse monoclonal anti-acetylated tubulin (Sigma, [6-11B-1], T7451) with Abberior Star 635P goat anti-mouse IgG (H + L) (Abberior GmbH ST635P-1001–500 UG), rat monoclonal anti-tyrosinated tubulin (Abcam, [YL1/2], ab6160) with Alexa Fluor 594 goat anti-rat IgG (H + L) (Molecular Probes, Life Technologies A11007) and rabbit recombinant anti-alpha tubulin antibody (Abcam, [EP1332Y], 52866), rabbit polyclonal anti-detyrosinated tubulin antibody (Merck, AB3210), and rabbit polyclonal anti-delta2 tubulin antibody (Millipore, AB3203) with Alexa Fluor 488 goat anti-rabbit IgG (H+L) (Thermo Fisher Scientific, A-11034). For staining of COS7 cells the antibody combinations were slightly different to ensure optimal signal intensity. We used rabbit polyclonal anti-detyrosinated tubulin antibody with Abberior Star 635P goat anti-rabbit igG (H+L)(Abberior GmbH ST635P-1002–500 UG), mouse monoclonal anti-acetylated tubulin with Alexa Fluor 594 goat anti-mouse IgG (H+L)(Molecular Probes, Life Technologies A11032) and rat monoclonal anti-tyrosinated tubulin with Alexa Fluor 488 goat anti-rat IgG (H+L)(Molecular Probes, Life Technologies A11006).

## Expansion microscopy

Expansion microscopy (ExM) was performed according to the proExM protocol (*Tillberg et al., 2016*) with the detailed description published in *Jurriens et al., 2021*. Briefly, immunostained neurons on 18-mm glass coverslips were incubated overnight in PBS with 0.1 mg/ml Acryloyl-X (Thermo Fisher, A20770) and afterwards washed three times with PBS. Per coverslip, we made 200 µl of gelation solution by mixing 188 µl of monomer stock solution (1 × PBS, 2 M NaCl, 8.625% (w/w) sodium acrylate (SA)(Sigma Aldrich 408220), 2.5% (w/w) acrylamide (AA), 0.15% (w/w) N,N'-methylenebisacrylamide), 8 µl of 10% (w/w) tetramethylethylenediamine (TEMED, BioRad 161–0800) accelerator and 4 µl of 10% (w/w) ammonium persulfate (APS, Sigma Aldrich A3678) initiator (added at the last step). Of the gelation solution, 120 µl was transferred to a gelation chamber, made out of a silicone mold with an inner diameter of 13 mm (Sigma-Aldrich, GBL664107) attached to a parafilm-covered glass slide. The sample was put cells-down on top of the chamber to close it off. After incubation at RT for 1–3 min, the sample was transferred to a humidified 37˚C incubator for at least 30 min to fully polymerize the gel. After gelation, the gel was transferred to a 12-well plate with 2 ml of digestion buffer (1 × TAE buffer (40 mM Tris, 20 mM acetic acid, 1 mM EDTA, pH8), 0.5% Triton X-100, 800 mM NaCl, 8 U/mL proteinase-K (ThermoFisher, EO0492)) for 4 hr at 37˚C for digestion. The gel was transferred to 50 ml deionized water for overnight expansion, and water was refreshed once to ensure the expansion reached plateau. Plasma-cleaned 24 × 50 mm rectangular coverslips (VWR 631–0146) for gel imaging were incubated with 0.1% poly-l-lysine to reduce drift of the gel during acquisition. The gel was mounted using custom-printed imaging chambers (*Jurriens et al., 2021*). The expansion factor was calculated for each sample as a ratio of a gel's diameter to the diameter of the gelation chamber and was in the range of 4.14–4.16. For the FlipExM samples, we cut a thin piece of gel (1 cm x 3 mm) using a razor blade and flipped it on its cut edge during transfer to the imaging chamber.

## STED imaging

Data from non-expanded samples were acquired using a Leica TCS SP8 STED 3X microscope with a pulsed (80MHz) white-light laser, HyD detectors and spectroscopic detection using HC PL APO 100×/1.40 Oil STED WHITE (Leica 15506378) oil-immersion objective. For Abberior STAR 635P and Alexa 594 we used 633 nm and 594 nm laser lines for excitation and a 775 nm synchronized pulsed laser for depletion, with a time gating range of 0.3–7 ns. For Alexa 488 we used 488 nm excitation, 592 nm continuous depletion laser line and time gate of 1.1–7 ns. Emission detection windows were

500–560 nm, 605–630 nm and 640–750 nm for Alexa 488, Alexa 594 and Abberior STAR 635P, respectively. No bleed-through was observed between the channels. For three-color cell body imaging (*Figure 2*, *Figure 3*), each fluorescent channel was imaged using the 2D STED configuration (vortex phase mask) in sequential z-stack mode from highest to lower wavelength, to prevent photobleaching by the 592 nm depletion laser line. For two-color imaging of dendrites (*Figure 1*), we used the Abberior STAR 635P/Alexa 594 combination and a single 775 nm depletion line and therefore acquired images in line-sequential mode. For the 3D STED imaging, we used a combined depletion PSF light path consisting of a mixture 60% Z-donut and 40% vortex phase mask, providing approximately isotropic resolution.

For the data shown in *Figure 2*, *Figure 3*, the size of the field-of-view was in the range of 30–50 μm and it was positioned to include the whole cell body of a neuron (soma) and the first 5–10 μm of dendrites emanating from it (*Figure 2A*, *Figure 3A*). For the data shown in *Figure 1*, the size of the field-of-view was in the range of 50–100 μm and it covered 30–50 μm of the proximal dendrites. The depth of z-stacks varied in the range from 3 to 6 μm for each acquisition and for all cases it was chosen to fully cover the dendrite's thickness. The lateral pixel size was in the range of 27–30 nm with a distance between z-planes in the range of 150–160 nm. The z-stacks were subjected to a mild deconvolution using Huygens Professional software version 17.04 (Scientific Volume Imaging, The Netherlands) with CMLE (classic maximum likelihood estimation) algorithm with parameters of SNR (Signal-to-Noise Ratio) equal to 7 over 10 iterations. After the deconvolution, z-stacks of tyrosinated and acetylated channels were registered in 3D to total tubulin channel using maximum intensity projections in XY and XZ planes using Correlescence plugin v.0.0.4 (https://github.com/ekatrukha/Correlescence archived on Zenodo repository https://doi.org/10.5281/zenodo.4534715) for ImageJ.

## ExM/FlipExM samples imaging

Expanded gels were imaged using the same Leica TCS SP8 STED 3X microscope with a pulsed (80 MHz) white-light laser, HyD detectors and spectroscopic detection using a HC PL APO 86 ×/1.20 W motCORR STED (Leica 15506333) water-immersion objective with a correction collar. Each fluorescent channel was imaged in confocal line-sequential mode. For Alexa488, we used 488 nm excitation and 500–560 nm emission range, for Alexa594 we used 594 nm excitation and 605–630 nm emission and Abberior STAR 635P we used 633 nm excitation and 640–750 nm emission. For ExM samples, the size of the field-of-view was in the range of 50–100 μm and had a thickness in the range of 10–20 μm, chosen to cover the whole volume of a dendrite. The dimensions of FlipExM stacks were 20–30 μm in XY and 30–50 μm in Z. In both cases, the pixel size in XY plane was in the range of 60–80 nm and the distance between z-planes was in the range of 150–180 nm. The z-stacks were subjected to a mild deconvolution using Huygens Professional software version 17.10 (Scientific Volume Imaging, The Netherlands) with CMLE (classic maximum likelihood estimation) algorithm with parameters of SNR (Signal-to-Noise Ratio) equal to 15 over 10 iterations.

## Single microtubule intensity estimate in the cell body using STED

From registered z-stacks, we chose substacks of 4–6 frames (0.7–1 μm thick) located at the bottom of the cell under the nucleus (*Figure 2A*), where the density of microtubule network was low. Using maximum intensity projections of these substacks in each fluorescent channel, we extracted segments of microtubule filaments using CurveTrace plugin ver.0.3.5 (https://github.com/ekatrukha/CurveTrace archived on Zenodo repository https://doi.org/10.5281/zenodo.4534721) for ImageJ implementing (*Steger, 1998*). The detection parameters used were: line width of 2.5 pixels (75 nm) (standard deviation of line thickness) and minimum segment's length of 0.6 μm. The detection of filaments was limited to the area of cell body, excluding dendrites (*Figure 2B*, *Figure 2—figure supplement 1*). After detection, each segment of microtubule was stored as a polyline ROI (region of interest) file in ImageJ format, essentially represented as a set of ordered XY coordinates. The detection was performed separately for each fluorescent channel, to take advantage of sparser filament's subnetworks with less overlap, displayed in tyrosinated and acetylated channel (*Figure 2B*, bottom row).

The quantification of filament intensities was performed on the sum of slices of the substacks used for the detection (SUM projection). The intensity of filament segments detected in the different channels was quantified for each fluorescent channel (total, tyrosinated, acetylated), producing nine

datasets (*Figure 2C*). For each detected polyline ROI segment of length $L_{\mathrm{mid}}$, we first measured the integrated intensity $I_{\mathrm{mid}}$ with a line width $w_{\mathrm{microtubule}}$ of 6 pixels (~180 nm) covering the area of $S_{\mathrm{mid}}$. In this case $w_{\mathrm{microtubule}}$ corresponds to the width used to normalize intensity of a single microtubule segment. As a second step, we measured the integrated intensity $I_{\mathrm{wide}}$ of the same segment with a wider line's width of 13 pixels (~390 nm) covering the area of $S_{\mathrm{wide}}$. From these measurements, the average intensity of the background $I_{\mathrm{BG}}$ was calculated as:

$$I_{\mathrm{BG}} = \frac{I_{\mathrm{wide}} - I_{\mathrm{mid}}}{S_{\mathrm{wide}} - S_{\mathrm{mid}}} \tag{1}$$

and the background-corrected average intensity of a segment $I_{\mathrm{segm}}$ as:

$$I_{\mathrm{segm}} = \frac{I_{\mathrm{mid}} - I_{\mathrm{BG}} S_{\mathrm{mid}}}{S_{\mathrm{mid}}} \tag{2}$$

Since the line width was fixed, this value does not depend on the length of the filament and essentially represents average fluorescent intensity along the filament. The described measurements were automated using an ImageJ script (*Katrukha, 2021*).

The histogram of segment intensities in total channel (pooled from all three detections) was fitted with a sum of two Gaussians (*Figure 2E*) expressed as:

$$\rho_{\mathrm{segm}}\left(I_{\mathrm{segm}}\right) = a_1 \exp\left(\frac{-\left(I_{\mathrm{segm}} - I_{\mathrm{Tot}}\right)^2}{2\sigma_{\mathrm{Tot}}^2}\right) + a_2 \exp\left(\frac{-\left(I_{\mathrm{segm}} - 2I_{\mathrm{Tot}}\right)^2}{4\sigma_{\mathrm{Tot}}^2}\right) \tag{3}$$

where $a_1$, $a_2$ correspond to the amplitudes (weights) of first and second Gaussians, $I_{\mathrm{Tot}}$ and $\sigma_{\mathrm{Tot}}$ are the average intensity and standard deviation of the Gaussian corresponding to the single microtubule intensity distribution (for the second Gaussian, after the convolution, average intensity and standard deviation are $2I_{\mathrm{Tot}}$ and $\sqrt{2}\sigma_{\mathrm{Tot}}$). The fitting was performed for each cell individually, to eliminate a difference in imaging conditions and heterogeneity of the sample.

The fitted value of average intensity $I_{\mathrm{Tot}}$ was used later for the estimation of total microtubule numbers in dendrites (see next section). For quantification of the average levels of tyrosination and acetylation per single microtubule, we introduced a threshold of $I_{\mathrm{Tot}} + \sigma_{\mathrm{Tot}}$ on the corresponding total tubulin intensity of segments detected in acetylated and tyrosinated channels (*Figure 2E*). Only the segments which total tubulin intensity was below this threshold were used for the calculation of average intensities of single tyrosinated $I_{\mathrm{Tyr}}$ or acetylated $I_{\mathrm{Ac}}$ microtubules. The average values for each channel were used for the normalization of intensities presented at *Figure 2G,H* and *Figure 2— figure supplment 2*. The fitting and threshold filtering was performed using custom written MATLAB scripts (*Katrukha, 2021*).

## Estimation of microtubules number in dendrites using STED

To estimate the average number of microtubules in dendrites, we first built summary (integrated) XY projection images of the z-stacks containing the whole depth of dendrites. Similar to the quantification of single microtubule intensity, we drew a straight line ROI of 2-3 μm ($L_{\mathrm{dendrite}}$) along a dendrite segment with a width $w_{\mathrm{dendrite}}$ (using ImageJ). This width varied depending on the dendrite and was chosen to visually include its whole thickness, covering an area of $S_{\mathrm{mid}}^{\mathrm{dendrite}}$. We measured the integrated intensity over this area, denoted as $I_{\mathrm{mid}}^{\mathrm{dendrite}}$. In the second step, we measured the integrated intensity $I_{\mathrm{wide}}^{\mathrm{dendrite}}$ of the same straight line with a width increased by 10 pixels (300 nm) covering the area of $S_{\mathrm{wide}}^{\mathrm{dendrite}}$. From these measurements, the average intensity of the background $I_{\mathrm{BG}}^{\mathrm{dendrite}}$ was calculated similar to *Equation (1)* as:

$$I_{\mathrm{BG}}^{\mathrm{dendrite}} = \frac{I_{\mathrm{wide}}^{\mathrm{dendrite}} - I_{\mathrm{mid}}^{\mathrm{dendrite}}}{S_{\mathrm{wide}}^{\mathrm{dendrite}} - S_{\mathrm{mid}}^{\mathrm{dendrite}}} \tag{4}$$

and the background-corrected average intensity per area of a dendrite segment $I^{\mathrm{dendrite}}$ was calculated as:

$$I^{\mathrm{dendrite}} = \left(I_{\mathrm{mid}}^{\mathrm{dendrite}} - I_{\mathrm{BG}}^{\mathrm{dendrite}} S_{\mathrm{mid}}^{\mathrm{dendrite}}\right) / \left(L_{\mathrm{dendrite}} w_{\mathrm{microtubule}}\right) \tag{5}$$

$$w_{\text{microtubule}}$$

$$n_{\text{Tot}} = \frac{I_{\text{Tot}}^{\text{dendrite}}}{I_{\text{Tot}}} \qquad (6)$$

where $I_{\text{Tot}}^{\text{dendrite}}$ is dendrite's intensity in total tubulin channel calculated according to *Equation (5)* and $I_{\text{Tot}}$ is average single microtubule intensity in total tubulin channel (see previous section). The specific values of $I_{\text{Tot}}$ were taken from the same cell/z-stack containing the dendrite.

To calculate numbers of microtubules in tyrosinated and acetylated channel we used following formulas (*Figure 2I*):

$$I_{\text{Tyr}}^{\text{dendrite}} = \left(n_{\text{Tyr}} + \alpha n_{\text{Ac}}\right) I_{\text{Tyr}} \qquad (7)$$

$$I_{\text{Ac}}^{\text{dendrite}} = \left(\beta n_{\text{Tyr}} + n_{\text{Ac}}\right) I_{\text{Ac}} \qquad (8)$$

where $I_{\text{Tyr}}^{\text{dendrite}}$, $I_{\text{Ac}}^{\text{dendrite}}$ are background corrected dendrite intensities calculated according to *Equation (5)*, $I_{\text{Tyr}}$ and $I_{\text{Ac}}$ average single microtubule intensities in tyrosinated and acetylated channel, $\alpha$ stands for average level of tyrosination for microtubules detected in the acetylated channel, $\beta$ corresponds to the average acetylation level of microtubules detected in the tyrosinated channel (*Figure 2H*) and $n_{\text{Tyr}}, n_{\text{Ac}}$ are numbers of tyrosinated and acetylated microtubules. The solution of system *Equation (7)-(8)* gives the final formulas:

$$n_{\text{Tyr}} = \frac{\theta_{\text{Tyr}} - \alpha \theta_{\text{Ac}}}{(1 - \alpha\beta)} \qquad (9)$$

$$n_{\text{Ac}} = \theta_{\text{Ac}} - \beta n_{\text{Tyr}} \qquad (10)$$

where $\theta_{\text{Tyr}} = I_{\text{Tyr}}^{\text{dendrite}}/I_{\text{Tyr}}$ and $\theta_{\text{Ac}} = I_{\text{Ac}}^{\text{dendrite}}/I_{\text{Ac}}$. *Equations (9)-(10)* were used to report the number of tyrosinated and acetylated microtubules in *Figure 3C-D*. In addition, these values were used to calculate the tyrosinated and acetylated percent of total microtubules number $n_{\text{Tot}}$ reported in *Figure 3E*. The number of 'non-modified', other microtubules was calculated as $n_{\text{Other}} = n_{\text{Tot}} - n_{\text{Tyr}} - n_{\text{Ac}}$.

The dendritic cross-section area (*Figure 3C–D*) was measured by building XZ resliced cross-section along a perpendicular line in the area of intensity measurement.

## Radial distribution of intensities in dendrites

We acquired z-stacks covering the whole thickness of a dendrite (using 2D or 3D STED in *Figure 1* and confocal for ExM samples in *Figure 3*). Using maximum intensity projection in XY plane, we marked the middle of dendrite with a polyline ROI of appropriate thickness. We used 'Selection->Straighten' function of ImageJ on original z-stacks to generate B-spline interpolated stacks, so a dendrite became straight and oriented along X axis. From those stacks, we generated a resliced stack in the plane perpendicular to dendrite's axis (YZ) for the analysis of radial intensity distribution in the cross-section (*Figure 1A–B*). To find the boundary outline of the dendrite in each slice, we used a custom written set of ImageJ macros allowing semi-automated analysis (*Katrukha, 2021*). The process consisted of two stages: finding the bounding rectangle encompassing the dendrite's intensity and building a smooth closed spline (approximately in the shape of an oval, see below). Illustration of full analysis workflow is presented in *Video 1*.

First, we calculated a center of mass (based on intensity) coordinates $x_c$, $y_c$ for tyrosinated (STED data) or total tubulin (ExM) channels. Then we specified a rectangular ROI *R* of maximum area under conditions that it was still located inside the image and that its center (intersection of diagonals) was positioned at the center of mass. In the next step, we progressively downsized the rectangle from each side to find the position where edge's intensity becomes equal to some threshold value (see below). We describe it here for the right side, but the same procedure was applied to all sides. Given an initial rectangle *R* of width *w*, height *h* and top left corner coordinates $x_R$, $y_R$, we built a set of rectangles with the width $w_i$ ranging from *w*/2 to *w* (with a step size of one pixel) with the same

height $h$ and same and fixed position of the top left corner. For each rectangle from this set, we measured the integrated intensity, providing the integrated intensity as a function of width $I_{int}(w_i)$. The intensity of the edge $I_e(w_i)$ was calculated as a derivative of this function, that is $I_e(w_i)=I_{int}(w_i) - I_{int}(w_{i+1})$ and normalized by its maximum and minimum value. A typical shape of $I_e(w_i)$ represents a peak around $w/2$ (a center of dendrite/rectangle) that is gradually decaying toward periphery. For the first image in the resliced stack, by decreasing the value of $w_i$ starting from $w$, we found the first value of $I_e(w_i)$ that exceeds a threshold normalized intensity value of $I_{thr}$ and its corresponding width $w_{RB}$. The coordinate of the right boundary (RB) was calculated as $x_{RB}=x_R +w_{RB}$. The threshold intensity value $I_{thr}$ was in the range of 0.2–0.4 and chosen for each first image in a stack manually to provide the values of $x_{RB}$ corresponding to the visual boundary of dendrite's intensity. The procedure was repeated for all other sides of the rectangle $R$, providing coordinates of left $x_{LB}$, top $y_{TB}$ and bottom $y_{BB}$ boundaries. For horizontal boundaries, the width was kept the same and the position of the opposite edge was kept constant while building $I_{int}(h_i)$. Using the newly found coordinates of the boundaries, we thereby built updated rectangle $R_B$ encompassing dendrite's cross-section.

This method worked robustly in many cases, but it failed in the presence of axons that were often wrapped around a dendrite. In the YZ plane, those axons produced additional fluorescent spots next to dendrite cross-section that were included into rectangle $R_B$. In the shape of $I_e(w_i)$ curve they manifest themselves as additional local peaks. Therefore, procedure of finding $R_B$ from initial rectangle $R$ for all other images in the resliced YZ stack (apart from the first) was modified. In these cases, we scanned $I_e(w_i)$ by both decreasing and increasing values of $w_i$ in the $w/2$ to $w$ range. During a scan, we recorded all values of $w_i$ that corresponded to each threshold $I_{thr}$ from the set of 0.1 to 0.5 with a step of 0.1. From these we calculated a set of candidate right boundary positions, from which we chose the one that is closest to the corresponding boundary from the previous slice image in the stack. This value was recorded as the new edge of $R_B$ rectangle at the current image. The procedure was repeated for each edge and after finding boundary rectangles for the whole stack, they were inspected and corrected manually.

To build a closed smooth spline contour around the irregular shaped dendrite's cross-section, in addition to vertical and horizontal boundaries, we also determined diagonal boundary points. For that we built an intensity profile along the 20–40 pixels wide line ROI connecting left top and right bottom corners of the rectangle $R_B$. After normalization of intensity to minimum and maximum, we found coordinates of two points on the both halves of line where intensity is closest to 0.15–0.2 of its maximum value, denoted $(x_{LD1}, y_{LD1})$ and $(x_{LD2}, y_{LD2})$. The same procedure was performed on the diagonal segment connecting left bottom and top right corners of rectangle RB, providing points $(x_{RD1}, y_{LD1})$ and $(x_{RD2}, y_{RD2})$. The ordered set of eight points with coordinates $(x_{LB}, y_c)$, $(x_{LD1}, y_{LD1})$, $(x_c, y_{TB})$, $(x_{RD1}, y_{LD1})$, $(x_{RB}, y_c)$, $(x_{LD2}, y_{LD2})$, $(x_c, y_{BB})$, $(x_{RD2}, y_{RD2})$ was used to construct smooth closed spline boundary $C$ passing through all of them (ImageJ functions *makePolygon* and '*Fit Spline*'). The final outlines for each image were inspected visually and if necessary, corrected manually.

In addition, for some ExM data profiles with low background we used an alternative, faster algorithm to find the boundary. We detected a set of points representing local fluorescence intensity maxima in each YZ slice (corresponding to MTs cross-sections). Using this set, we built a convex hull and constructed a spline from it. Again, we manually checked and corrected generated outlines.

To build the radial intensity distribution, for each image we iteratively reduced the contour $C$ with steps of one pixel using ImageJ function 'Enlarge ROI' with negative values (it uses Euclidean distance map threshold), while measuring its area $S_k$ and integrated intensity $IC_k$ (where index $k$ denotes the step). We calculated the average intensity $MI_k$ of each contour in the shrinking series as derivative:

$$MI_k = \frac{IC_k - IC_{k-1}}{S_k - S_{k-1}} \tag{11}$$

To get the radial distribution, for each $k$ step we recalculated area $S_k$ to radius using the formula $R_k = \sqrt{S_k/\pi}$ and normalized it by maximum value. Finally, to get a probability density function $\rho(R)$, we normalized $MI(R)$ by the area under the curve.

## Decomposition of radial intensities

For the decomposition of total tubulin radial density $\rho_{\mathrm{Tot}}(R)$ as a weighted sum of tyrosinated $\rho_{\mathrm{Tyr}}(R)$ and acetylated $\rho_{\mathrm{Ac}}(R)$ densities (*Figure 4G*) we minimized the mean square error MSE($w_{\mathrm{Tyr}}$, $w_{\mathrm{Ac}}$) between two curves:

$$MSE\left(w_{\mathrm{Tyr}}, w_{\mathrm{Ac}}\right) = \sum_{R} \left(\rho_{\mathrm{Tot}}(R) - w_{\mathrm{Tyr}}\rho_{\mathrm{Tyr}}(R) - w_{\mathrm{Ac}}\rho_{\mathrm{Ac}}(R)\right)^2 \tag{12}$$

where $w_{\mathrm{Tyr}}$ and $w_{\mathrm{Ac}}$ correspond to the weights of tyrosinated and acetylated densities. By taking the derivatives of *Equation (12)* and making them equal to zero, the solution can be found in a closed form:

$$w_{\mathrm{Ac}} = \frac{\langle\rho_{\mathrm{Tot}}\rho_{\mathrm{Ac}}\rangle\langle\rho_{\mathrm{Tyr}}^2\rangle - \langle\rho_{\mathrm{Tot}}\rho_{\mathrm{Tyr}}\rangle\langle\rho_{\mathrm{Tyr}}\rho_{\mathrm{Ac}}\rangle}{\langle\rho_{\mathrm{Tyr}}^2\rangle\langle\rho_{\mathrm{Ac}}^2\rangle - \langle\rho_{\mathrm{Tyr}}\rho_{\mathrm{Ac}}\rangle^2} \tag{13}$$

$$w_{\mathrm{Tyr}} = \frac{\langle\rho_{\mathrm{Tot}}\rho_{\mathrm{Tyr}}\rangle - w_{\mathrm{Ac}}\langle\rho_{\mathrm{Tyr}}\rho_{\mathrm{Ac}}\rangle}{\langle\rho_{\mathrm{Tyr}}^2\rangle} \tag{14}$$

where angle brackets denote averaging over the whole radius range. It must be noted, that even without addition of a stronger assumption $w_{\mathrm{Tyr}} + w_{\mathrm{Ac}} = 1$, our analysis provided values that satisfy this relation.

## Single microtubule intensity estimate in FlipExM YZ stacks

The cross-sections of microtubules in YZ FlipExM appeared as a set of fluorescent spots (*Figure 5B*). For intensity analysis, we used ComDet v.0.5.3 plugin for ImageJ (https://github.com/ekatrukha/ComDet archived on Zenodo repository https://doi.org/10.5281/zenodo.4281064), which reports spot area, width, height in the detected channel and quantifies the background corrected integrated intensity in all three channels. To filter out possible microtubule bundles, we applied a lower bound threshold of 0.8 on the spot 'roundness' $\theta$, expressed as:

$$\theta = \frac{w_{\mathrm{spot}}h_{\mathrm{spot}}}{max\left(w_{\mathrm{spot}}, h_{\mathrm{spot}}\right)^2} \tag{15}$$

where $w_{\mathrm{spot}}$ and $h_{\mathrm{spot}}$ correspond to spot width and height. For spot areas detected per channel and per dendrite, we performed an MLE fit to the normal distribution and obtained estimates for the mean $S_{\mathrm{mean}}$ and standard deviation $\sigma_{\mathrm{area}}$, which were used to filter out spots with areas outside the inclusion range with lower bound max($Q_1$, $S_{\mathrm{mean}} - \sigma_{\mathrm{area}}$) and upper bound ($S_{\mathrm{mean}} + \sigma_{\mathrm{area}}$), see *Figure 5C,D*. For the lower bound, $Q_1$ stands for 25th percentile, it was added to robustly remove false positives.

After the 'roundness' and area filters, an estimation of average single microtubule intensity was performed in a similar way as in *Figure 2*, that is by fitting a sum of two Gaussians (*Equation (3)*) to the histogram of intensity distributions (*Figure 5E*, *Figure 5—figure supplement 1*). For each slice of YZ stack, we calculated normalized integrated intensity in each channel as a sum of all spot's intensities divided by a single microtubule intensity derived from the fit. To calculate absolute MTs numbers per slice, we used the same *Equation (6)-(8)* as in *Figure 1*. Since there was little variability in the MTs numbers along the proximal part of the dendrite used for analysis (*Figure 5—figure supplement 4*), we calculated average MT number over all slices for each channel per dendrite, reported in *Figure 5H*.

To calculate the average level of tyrosination of microtubules detected in the acetylated channel $\alpha$ and the average level of acetylation of microtubules detected in the tyrosinated channel $\beta$ from the FlipExM data, we used three different methods. Here, the estimation was more challenging, because their distributions displayed very long tails and therefore the average crosstalk values were still quite large (*Figure 5—figure supplement 1A*). In the first method, we estimated $\alpha$ and $\beta$ by pooling all filtered spot detections and calculating their average values (*Figure 5F,G*; *Figure 5—figure supplement 5*, left panels). In the second method, we only included intensities in the acetylated/

tyrosinated detection channels that were less than the values of mean + SD in the same channel (*Figure 5—figure supplement 5*, middle panels). In the third method, we fitted the distribution of acetylation levels on tyrosinated MTs (and vice versa) with a sum of two Gaussian function, to obtain the values of $\alpha$ and $\beta$ as positions of peaks at the Ace/Tyr plane (*Figure 5—figure supplement 5*, right panels). The two last methods provided smaller values for $\alpha$ and $\beta$, which resulted in slightly different estimates for the percentage of tyrosinated/acetylated microtubules (*Figure 5—figure supplement 5*).

### STED resolution

The lateral resolution of STED 2D (*Figure 1—figure supplement 2A*, x-axis) was calculated using parameter-free decorrelation method (*Descloux et al., 2019*) on the maximum intensity projection of cell body z-stacks (*Figure 2B*). Same images and MT segments detections were used to calculate an average FWHM of individual MTs segments (*Figure 1—figure supplement 2A*, y-axis) using 'Fit Gaussian to Curves' function of CurveTrace ImageJ plugin (see above). In addition, vertical and horizontal Gaussian fits to the YZ-cross-sections images of MTs from (*Figure 1A–B*, *Figure 2B*) were used to estimate lateral and axial resolution of STED 2D and 3D (*Figure 1—figure supplement 2B*).

### Intensity analysis along a single dendrite

For the analysis of MTs modifications along the length of individual dendrites (*Figure 1—figure supplement 3*), we performed a series of confocal tilescan acquisitions of large areas (100–400 µm in size) around neuron cell bodies. Each tile was acquired as a z-stack covering the whole thickness of dendrites with the same excitation/emission settings as described in 'ExM/FlipExM samples imaging' section. Tiles' SUM projections were stitched together in ImageJ using 'Pairwise Stitching' plugin (*Preibisch et al., 2009*). Individual dendrites were traced manually with polyline ROI in ImageJ, which was subsequently fitted with a spline. We used a custom written ImageJ macro to fit perpendicular intensity profile at equidistantly sampled points of the ROI in each channel to a Gaussian function with a background offset. The fluorescent intensity at each point along a dendrite was estimated as multiplication of amplitude to the standard deviation of fitted Gaussian. Using Matlab script, the intensity profiles at each channel were normalized by the average value of first 5 µm and smoothened with a window of 2 µm. Intensity values above 1.25 were excluded from the analysis to remove occasional intensity spikes caused by intersecting neurites. The values of fitted standard deviation in total tubulin channel was used to calculate FWHM (*Figure 1—figure supplement 3*).

## Acknowledgements

This work was supported by the European Research Council (ERC Consolidator Grant 819219) and ZonMW (project 91217002).

## Additional information

### Funding

| Funder | Grant reference number | Author |
| --- | --- | --- |
| H2020 European Research Council | 819219 | Lukas C Kapitein |
| ZonMw | 91217002 | Daphne Jurriens<br>Lukas C Kapitein |

The funders had no role in study design, data collection and interpretation, or the decision to submit the work for publication.

### Author contributions

Eugene A Katrukha, Data curation, Software, Formal analysis, Investigation, Visualization, Methodology, Writing - original draft; Daphne Jurriens, Investigation, Methodology, Writing - original draft; Desiree M Salas Pastene, Investigation; Lukas C Kapitein, Conceptualization, Supervision, Funding acquisition, Methodology, Writing - review and editing

## Author ORCIDs

Eugene A Katrukha (iD) https://orcid.org/0000-0002-9971-3603
Daphne Jurriens (iD) https://orcid.org/0000-0001-5123-3099
Lukas C Kapitein (iD) https://orcid.org/0000-0001-9418-6739

## Ethics

Animal experimentation: Culturing of neurons has been approved by the ethical commitee (DEC) of Utrecht University and by the Centrale Commissie Dierproeven of the Dutch government (permit application AVD1080020173404). The ethical committee (DEC) is independent and must review any experimental use of animals in the Netherlands.

## Decision letter and Author response

Decision letter https://doi.org/10.7554/eLife.67925.sa1
Author response https://doi.org/10.7554/eLife.67925.sa2

# Additional files

## Supplementary files

• Transparent reporting form

## Data availability

All quantitative data is available on Figshare: https://doi.org/10.6084/m9.figshare.c.5306546.v3. Software is available on Zenodo: https://doi.org/10.5281/zenodo.4281064 https://doi.org/10.5281/zenodo.4534715 https://doi.org/10.5281/zenodo.4534721.

The following datasets were generated:

| Author(s) | Year | Dataset title | Dataset URL | Database and Identifier |
|---|---|---|---|---|
| Katrukha EA, Jurriens D, Salas Pastene DM, Kapitein LC | 2021 | Quantitative mapping of dense microtubule arrays in mammalian neurons | https://doi.org/10.6084/m9.figshare.c.5306546.v3 | figshare, 10.6084/m9.figshare.c.5306546.v3 |
| Katrukha EA | 2021 | ekatrukha/ComDet: ComDet 0.5.3 | https://doi.org/10.5281/zenodo.4281064 | Zenodo, 10.5281/zenodo.4281064 |
| Katrukha EA | 2021 | ekatrukha/Correlescence v0.0.4 | https://doi.org/10.5281/zenodo.4534715 | Zenodo, 10.5281/zenodo.4534715 |
| Katrukha EA | 2021 | ekatrukha/CurveTrace v0.3.5 | https://doi.org/10.5281/zenodo.4534721 | Zenodo, 10.5281/zenodo.4534721 |

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
