## [Decision Letter]

**Acceptance summary:**

This paper utilizes advanced microscopy techniques to map the distribution of different microtubule populations within the dendrites of neurons. This previously posed a challenge to the field due to the tight bundling of microtubules along neuronal processes, but the authors elegantly overcome this challenge by measuring the density of microtubules by two independent methods to determine the proportion of acetylated and tyrosinated microtubules along the dendrite shaft. This work opens new avenues for exploring the basis of molecular motor transport within the complex polarized environment of the neuron.

**Decision letter after peer review:**

Thank you for submitting your article "Quantitative mapping of dense microtubule arrays in mammalian neurons" for consideration by *eLife*. Your article has been reviewed by 3 peer reviewers, and the evaluation has been overseen by a Reviewing Editor and Suzanne Pfeffer as the Senior Editor. The following individual involved in review of your submission has agreed to reveal their identity: Ilaria Testa (Reviewer #3).

Essential Revisions:

Overall, the reviewers agree this is an excellent paper that warrants publication in *eLife* after a few revisions.

1) Reviewers 1 and 2 asked about the limits in distinguishing individual microtubules in smaller processes, such as distal dendrites and axons. If possible, can the authors provide data on the subsets of modified microtubules along the entire length of a dendrite, or at multiple sections closer to the proximal end? Or provide an explanation as to the limits precluding this analysis? Along these lines, are the authors able to measure the spacing of microtubules to determine if microtubule density varies with dendrite diameter along a single dendrite?

2) There were a few comments on antibodies by Reviewers 2 and 3. Are the authors able to show that they can detect the non-tyrosinated microtubules with an antibody for detyrosination or delta2-tubulin? If not, the authors should still mention in the manuscript that the co-staining of the acetylated microtubule population with either of these two antibodies would be expected. This discussion could also highlight the limitations of the currently available antibodies to tubulin PTMs.

3) Is it possible to estimate microtubule length? For example, as Reviewer 1 suggests: to label the microtubule plus- and minus-ends for imaging with STED and/or ExM? Or is there an alternative way to add this parameter into your manuscript?

4) Please respond to this comment from Reviewer 3: The total fitted curve (gray) in Figure 5E does not seem to match the filtered data points (black dots) so well at larger intensity values (like it was doing in the soma data). I suspect this could be due to bundles with a higher number of tubules inside, which might be more abundant in the dendrites than in the soma? There is basically just one point of confusion for me, and that is why they get similar α and β factors in Figure 2H and Figure 5G, despite the distributions in the 2D heat map being more separated and aligned horizontally and vertically in 5G. I could be missing something, but I thought those factors would be higher the more blurred together the two groups are in the heatmap. The author wrote: "The distributions of tyrosinated and acetylated detections in the tyrosination/acetylation plane (Figure 5F, S4C,D) appeared very similar to the data obtained earlier using the cell body (Figure 2G), but with more distinct separation between the two clusters. Compared to the cell body data, we found slightly different values for the average tyrosination level of acetylated microtubules (0.45 {plus minus} 0.05), as well as the acetylation level for tyrosinated (0.60 {plus minus} 0.17) (Figure 5G)" These two last sentences seem to contradict each other; if the clusters are more separated I would assume also the average "crosstalk" levels would be smaller, but maybe I am missing something here.

5) Please provide an estimation of the resolution of the 2D and 3D STED systems in the different channels.

Although these are the major points, please also refer to the individual reviewer comments for important changes to the text. The reviewers asked for further discussion on integrating your group's past results with these results, as well as certain choices in methodology and/or analysis and the limitations of the presented techniques.

*Reviewer #1 (Recommendations for the authors):*

– The work focuses on the proximal, thick part of dendrites. What happens in smaller processes, such as distal dendrites and axons? Where is the limit to distinguishing individual microtubules (notably for the most precise flip-ExM method)?

– Pre-labeling ExM (that is, expanding after immunolabeling) only inserts void between antibodies – could post-labeling ExM (immunolabeling after expansion) allow for better delineation of individual microtubules in tight bundles?

– How does microtubule density vary with dendrite diameter along a single dendrite? Are microtubules closer to each other when a dendrite becomes thinner, or do some of them stop, with a constant density?

– Microtubule length is a very important parameter given the role of MTs as rails for transport. Would it be possible to label microtubule extremities (plus- and minus-ends) for imaging with STED and/or ExM? Together with the obtained density estimates, this would allow for a statistical estimation of MT length.

– Another important aspect is the orientation and its correlation with modification/localization. This is only briefly mentioned (Discussion p. 9) without mentioning the related results in the author's Tas et al., Neuron 2018, ref 17. Would a precise estimation of MT extremities density also allow to derive their average orientation depending on the localization (along a dendrite and within transverse sections?).

– Finally, an estimation of MT length would also allow assessing if the same microtubules have different domains bearing distinct modification (as shown in Baas et al., JCB 1990).

*Reviewer #2 (Recommendations for the authors):*

1) In the abstract, the authors talk about "stable, acetylated" and "dynamic, tyrosinated" microtubules. While this is the common view in the field, there is no experimental evidence in this manuscript that the microtubules they observed are really stable when they are acetylated, and dynamic when tyrosinated. It might therefore be more appropriate to simply say that they observe acetylated and tyrosinated microtubules, and keep the suggestions that this reveals stable and dynamic microtubule fractions for the discussion? Or alternatively, use the term "presumed stable / dynamic microtubules"? In the introduction, the authors nicely explain the connection between these PTMs and why these microtubule populations are assumed to be dynamic or stable.

2) The paper omits to tell the reader why other tubulin PTMs have not been analysed with this method. This could have different reasons: other antibodies are less amenable for this method, other PTMs are less clearly segregated on different microtubule populations, etc. Given that the manuscript does a beautiful job in discussing all pros and cons of the presented work, it would be a great addition to tell the reader about the current limitations for other PTMs.

3) The authors must be more explicit about the fact that the method, despite considering different labelling intensities for different antibodies, still ends up binarizing the readout (they talk mostly about acetylated vs. tyrosinated microtubules). While this classification is insightful and important for the study of the cytoskeleton, it is important to avoid perpetuating the notion that individual microtubule identities related to single PTMs. The authors should be very clear that this is a threshold-based classification. A suggestion would be to be more explicit on this in the discussion, as it is clearly explained in the results.

4) Along the lines of point 1) and 2), the authors might consider discussing the limitation of the method to quantify PTMs that are intrinsically rheostatic, such as polyglutamylation.

5) The authors describe that filaments positive for acetylation show a decreased intensity for an α-tubulin antibody, interpreting that this might be due to decreased efficiency of the staining on this highly modified microtubule. A similar clash in compatibility might occur for acetylation and tyrosination staining (considering that both stainings are observed to different extent in both microtubule subsets). Can they exclude that this happens?

6) Following point 1) and 4), it would be a great addition to the current paper if the authors could show that they can detect the non-tyrosinated microtubules with an antibody for detyrosination. As it is likely that the authors have already tried this, another alternative could be delta2-tubulin. In neurons, long-lived microtubules often end up being delta2-positive following detyrosination. If none of this worked in the hands of the authors, they should still mention in the manuscript that the co-staining of the acetylated microtubule population with either of these two antibodies would be expected. This discussion could also highlight the limitations of the currently available antibodies to tubulin PTMs.

7) When imaging the dendrites, the authors select a region of length 5-10 μm proximal to the soma. They find that the ratios of mostly tyrosinated/mostly acetylated microtubules are very similar in both compartments. It would be informative to gain insight into the subsets of modified microtubules along the entire length of a dendrite, or at least at multiple sections closer to the proximal end. In Figure 1D and 4C the intensity distributions for tyrosination and acetylation suggest that shifts in the subsets distribution might occur further down the dendrite.

8) The newly developed FlipExM approach improves the resolution of single microtubules significantly, however, the method has the limitation of observing only a single cross section (1 cm x 3 mm) of the dendrite, on which the conclusions are based. The authors should explain how the specific regions of these cross sections were selected and controlled for the six different cells used for replicating the experiments (for example: which distance from the cell body was chosen?).

9) From Figure 5H and to some extent from Figure 3C it seems that with increasing diameter of the dendrite, and respectively microtubule number, the fraction of the acetylated filaments increases proportionally, while the peripheral tyrosinated and others fractions remain roughly the same. This may have an influence on the absolute numbers quantified and reported in this study about the percentage of the investigated modifications, and should be discussed.

10) The supplementary videos are very well done, and they help the reader to easily visualise the described approaches. Could the authors imagine to add a voice chancel explaining to the reader what is happening on the video? Or alternatively, make the explanatory text appear on the videos?

---

## [Author Response]

1) Reviewers 1 and 2 asked about the limits in distinguishing individual microtubules in smaller processes, such as distal dendrites and axons. If possible, can the authors provide data on the subsets of modified microtubules along the entire length of a dendrite, or at multiple sections closer to the proximal end? Or provide an explanation as to the limits precluding this analysis?

We focused on the detailed analysis of the proximal dendrites for various reasons. First, neuronal cultures are very dense and thin neurites from different cells often overlap and intertwine, which makes it challenging to separate and trace individual neurites further away from the cell body. Second, whole-dendrite imaging requires a tile-scan acquisition with overlaps between tiles, which in STED modality will lead to additional bleaching. Finally, when performing FlipExM, we use the cell bodies at the glass-gel interface as a reference point to identify dendrites. Since these dendrites are now aligned with the z-axis, this limits the dendritic length that we can image.

To address the request to provide data on the subsets of modified microtubules along the entire length of dendrites, we now performed confocal tile-scan acquisitions of large areas around neurons, including 50-100 µm long dendrites. We traced the intensity of total, tyrosinated and acetylated tubulin intensity along the dendrites (Figure 1—figure supplement 3) and found that both the relative acetylation and tyrosination levels slightly decrease along the length of a dendrite. Moreover, for the existing FlipExM data (Figure 5), we now provide the calculated number of microtubules within each subset as a function of dendrite length for the proximal part of the dendrite (Figure 5—figure supplement 4).

Along these lines, are the authors able to measure the spacing of microtubules to determine if microtubule density varies with dendrite diameter along a single dendrite?

Using the same whole-dendrite dataset introduced above we were able to measure total microtubule intensity as a function of dendrite thickness (estimated by calculating the full width at half maximum of a Gaussian fit to the intensity profile), as shown in Figure 1—figure supplement 3. We found that microtubule density displayed a quadratic dependence on dendrite diameter over a wide range of diameters (i.e. proportional to cross-section area). Only for the very thin dendrites (FWHM < 1µm) we observed a deviation from this dependence. This deviation most likely emerges from the much worse thickness estimate when approaching the diffraction limit. Thus, we do not see strong variations in microtubule density along single dendrites.

2) There were a few comments on antibodies by Reviewers 2 and 3. Are the authors able to show that they can detect the non-tyrosinated microtubules with an antibody for detyrosination or delta2-tubulin? If not, the authors should still mention in the manuscript that the co-staining of the acetylated microtubule population with either of these two antibodies would be expected. This discussion could also highlight the limitations of the currently available antibodies to tubulin PTMs.

To address this question, we tested the commercially available antibody against detyrosinated tubulin. While we were able to observe this MTs modification in cultured COS7 cells, we did not get reliable staining in neurons (Figure 1—figure supplement 4A,B). We furthermore tested the antibody against δ 2-tubulin and found that this modification was present on microtubule bundles that were positioned even more centrally comparing to the acetylated microtubules. We have added example images and the quantitative analysis to the revised manuscript (Figure 1—figure supplement 4C,D). However, since our focus is on acetylated and tyrosinated microtubules and little is known about the function of δ 2-tubulin, we decided to not pursue this further at this point.

3) Is it possible to estimate microtubule length? For example, as Reviewer 1 suggests: to label the microtubule plus- and minus-ends for imaging with STED and/or ExM? Or is there an alternative way to add this parameter into your manuscript?

We would indeed be very interested in measuring the length distribution of different microtubule subsets. Earlier work from the Shen lab indeed estimated microtubule length distribution in *C. elegans* axons by combining estimates for microtubule numbers with the distribution of a minus-end marker (Yogev et al., Neuron 2016). This was aided by the low overall microtubule numbers and the very clear and punctate distribution of PTRN-1, while also requiring addition assumptions about the most probable data interpretation. In our methodology, the usage of EBs or CAMSAPs as end markers is technically challenging, since antibodies for these proteins require methanol fixation, which is not compatible with the high-resolution imaging of microtubules using STED or ExM. In addition, it is not clear which fraction of microtubules will have EBs or CAMSAPs at their ends and how this differs for the different subsets. To address this, we have recently begun to establish CRISPR based knock-ins for CAMSAP proteins and have so far found that these proteins are not exclusively distributed in clear puncta at the ends of dendritic microtubules. Given these many uncertainties, we feel it would be premature to add such estimates to the current manuscript. We will continue to develop new techniques and labelling methods that will hopefully provide enough resolution to robustly trace individual microtubules and measure the length distribution of different subsets.

4) Please respond to this comment from Reviewer 3: The total fitted curve (gray) in Figure 5E does not seem to match the filtered data points (black dots) so well at larger intensity values (like it was doing in the soma data). I suspect this could be due to bundles with a higher number of tubules inside, which might be more abundant in the dendrites than in the soma? There is basically just one point of confusion for me, and that is why they get similar α and β factors in Figure 2H and Figure 5G, despite the distributions in the 2D heat map being more separated and aligned horizontally and vertically in 5G. I could be missing something, but I thought those factors would be higher the more blurred together the two groups are in the heatmap. The author wrote: "The distributions of tyrosinated and acetylated detections in the tyrosination/acetylation plane (Figure 5F, S4C,D) appeared very similar to the data obtained earlier using the cell body (Figure 2G), but with more distinct separation between the two clusters. Compared to the cell body data, we found slightly different values for the average tyrosination level of acetylated microtubules (0.45 {plus minus} 0.05), as well as the acetylation level for tyrosinated (0.60 {plus minus} 0.17) (Figure 5G)" These two last sentences seem to contradict each other; if the clusters are more separated I would assume also the average "crosstalk" levels would be smaller, but maybe I am missing something here.

For the FlipExM data, the estimation of α and β factors was more challenging for several reasons. First, there are more bundles with more than one microtubule, as correctly noted by the reviewer. Nonetheless, the separation between acetylation and tyrosination levels was indeed more apparent within dendrites (Figure 5F) and based on visual inspection, one would therefore expect lower values for α and β. However, these distributions have very long tails and therefore the average crosstalk values were still quite large. Because this could be a consequence of the inclusion of more bundled microtubules, we now include two alternative approaches to estimate α and β.

In the first approach, we only included intensities in the acetylated/tyrosinated detection channels that were less than the values of mean + SD in the same channel. (Figure 5—figure supplement 5, middle panels). In the second approach, we fitted the distribution of acetylation levels on tyrosinated MTs (and vice versa) with a sum of two Gaussian function, to obtain the values of α and β as positions of peaks at the Ace/Tyr plane (Figure 5—figure supplement 5, right panels). Both approaches indeed provided smaller values for α and β and resulted in slightly different estimates for the percentage of tyrosinated/acetylated microtubules (Figure 5—figure supplement 5B). We now show all these results in Figure 5—figure supplement 5, but kept the initial estimates in the main figures.

5) Please provide an estimation of the resolution of the 2D and 3D STED systems in the different channels.

To address this request, we have now estimated the resolution for 2D and 3D STED in two different ways by both decorrelation analysis [Descloux et al., Nat. Commun., 2019] and by analyzing the intensity profiles of individual microtubules. This revealed average lateral (axial) resolution of 110 (500) nm and 210 (310) nm for the 2D and 3D STED systems (we also report values for each channel). These results are presented in Figure 1—figure supplement 2.

Although these are the major points, please also refer to the individual reviewer comments for important changes to the text. The reviewers asked for further discussion on integrating your group's past results with these results, as well as certain choices in methodology and/or analysis and the limitations of the presented techniques.Reviewer #1 (Recommendations for the authors):– The work focuses on the proximal, thick part of dendrites. What happens in smaller processes, such as distal dendrites and axons? Where is the limit to distinguishing individual microtubules (notably for the most precise flip-ExM method)?

We focused on the detailed analysis of the proximal dendrites for various reasons. First, neuronal cultures are very dense and thin neurites from different cells often overlap and intertwine, which makes it challenging to separate and trace individual neurites further away from the cell body. Second, whole-dendrite imaging requires a tile-scan acquisition with overlaps between tiles, which in STED modality will lead to additional bleaching. Finally, when performing FlipExM, we use the cell bodies at the glass-gel interface as a reference point to identify dendrites. Since these dendrites are now aligned with the z-axis, this limits the dendritic length that we can image.

For the revised manuscript, we performed confocal tile-scan acquisitions of large areas around neurons, including 50-100 µm long dendrites. We traced the intensity of total, tyrosinated and acetylated tubulin intensity along the dendrites (Figure 1—figure supplement 3) and found that both the relative acetylation and tyrosination levels tend to slightly decrease along the length of a dendrite. Moreover, for the existing FlipExM data (Figure 5), we now provide the calculated number of microtubules within each subset as a function of dendrite length for the proximal part of the dendrite (Figure 5—figure supplement 4).

As also noted in the next comment, the fluorophore-epitope linkage error is not reduced when using conventional expansion microscopy. In earlier work, we have shown that this increases the full-width at half-maximum (FWHM) of microtubules to above 60 nanometers and precludes resolving bundled microtubules.

– Pre-labeling ExM (that is, expanding after immunolabeling) only inserts void between antibodies – could post-labeling ExM (immunolabeling after expansion) allow for better delineation of individual microtubules in tight bundles?

Indeed, post-expansion labeling should enable further resolution improvement. However, this is not trivial to implement, as it requires replacing proteolytic degradation by other ways of denaturation to preserve the epitopes. This is a direction that we want to explore in future work.

– How does microtubule density vary with dendrite diameter along a single dendrite? Are microtubules closer to each other when a dendrite becomes thinner, or do some of them stop, with a constant density?

In the revised manuscript, we included a whole-dendrite dataset for which we were able to measure the microtubule intensity as a function of dendrite thickness (Figure 1—figure supplement 3C). We found that microtubule density displayed a quadratic dependence on dendrite diameter over a wide range of diameters (i.e. proportional to cross-section area). Only for the very thin dendrites (FWHM < 1µm) we observed a deviation from this dependence. These results indicate that microtubule density remains fairly constant throughout the dendrite.

– Microtubule length is a very important parameter given the role of MTs as rails for transport. Would it be possible to label microtubule extremities (plus- and minus-ends) for imaging with STED and/or ExM? Together with the obtained density estimates, this would allow for a statistical estimation of MT length.

We would indeed be very interested in measuring the length distribution of different microtubule subsets. Earlier work from the Shen lab indeed estimated microtubule length distribution in *C. elegans* axons by combining estimates for microtubule numbers with the distribution of a minus-end marker (Yogev et al., Neuron 2016). This was aided by the low overall microtubule numbers and the very clear and punctate distribution of PTRN-1, while also requiring addition assumptions about the most probable data interpretation. In our methodology, the usage of EBs or CAMSAPs as end markers is technically challenging, since antibodies for these proteins require methanol fixation, which is not compatible with the high-resolution imaging of microtubules using STED or ExM. In addition, it is not clear which fraction of microtubules will have EBs or CAMSAPs at their ends and how this differs for the different subsets. To address this, we have recently begun to establish CRISPR based knock-ins for CAMSAP proteins and have so far found that these proteins are not exclusively distributed in clear puncta at the ends of dendritic microtubules. Given these many uncertainties, we feel it would be premature to add such estimates to the current manuscript. We will continue to develop new techniques and labelling methods that will hopefully provide enough resolution to robustly trace individual microtubules and measure the length distribution of different subsets.

– Another important aspect is the orientation and its correlation with modification/localization. This is only briefly mentioned (Discussion p. 9) without mentioning the related results in the author's Tas et al., Neuron 2018, ref 17. Would a precise estimation of MT extremities density also allow to derive their average orientation depending on the localization (along a dendrite and within transverse sections?).

We now cite our earlier work when discussion orientations. Unfortunately, we haven’t been able to develop robust ways to determine microtubule polarity in expanded samples. Motor-PAINT will obviously not work on expanded microtubules and is also challenging to use for 3D imaging in non-expanded thicker cells, due to the high number of fluorescent motors in solution.

– Finally, an estimation of MT length would also allow assessing if the same microtubules have different domains bearing distinct modification (as shown in Baas et al., JCB 1990).

We agree that robust imaging of microtubules along their entire length would enable us to address many new and important aspects of neuronal microtubule biology. The canonical view of neuronal microtubules featuring stable segments with dynamic ends (mostly based on data from axons) somehow contrasts our observation of distinct radial distributions of these subsets. Nonetheless, the distributions we observe are not completely mutually exclusive and different subsets can often be found located closely together, suggesting that future work might reveal such segmented microtubules.

Reviewer #2 (Recommendations for the authors):1) In the abstract, the authors talk about "stable, acetylated" and "dynamic, tyrosinated" microtubules. While this is the common view in the field, there is no experimental evidence in this manuscript that the microtubules they observed are really stable when they are acetylated, and dynamic when tyrosinated. It might therefore be more appropriate to simply say that they observe acetylated and tyrosinated microtubules, and keep the suggestions that this reveals stable and dynamic microtubule fractions for the discussion? Or alternatively, use the term "presumed stable / dynamic microtubules"? In the introduction, the authors nicely explain the connection between these PTMs and why these microtubule populations are assumed to be dynamic or stable.

We agree with the reviewer and removed the adjective ‘stable’ and ‘dynamic’ from the abstract.

2) The paper omits to tell the reader why other tubulin PTMs have not been analysed with this method. This could have different reasons: other antibodies are less amenable for this method, other PTMs are less clearly segregated on different microtubule populations, etc.. Given that the manuscript does a beautiful job in discussing all pros and cons of the presented work, it would be a great addition to tell the reader about the current limitations for other PTMs.

To address this question, we tested the commercially available antibody against detyrosinated tubulin. While we were able to observe this MTs modification in cultured COS7 cells, we did not get reliable staining in neurons (Figure 1—figure supplement 4A,B). We furthermore tested the antibody against δ 2-tubulin and found that this modification was present on microtubule bundles that were positioned even more centrally comparing to the acetylated microtubules. We have added example images and the quantitative analysis to the revised manuscript (Figure 1 figure supplement 4C,D). We furthermore added a statement on the current limitations for other PTMs.

3) The authors must be more explicit about the fact that the method, despite considering different labelling intensities for different antibodies, still ends up binarizing the readout (they talk mostly about acetylated vs. tyrosinated microtubules). While this classification is insightful and important for the study of the cytoskeleton, it is important to avoid perpetuating the notion that individual microtubule identities related to single PTMs. The authors should be very clear that this is a threshold-based classification. A suggestion would be to be more explicit on this in the discussion, as it is clearly explained in the results.

We indeed categorize microtubules as either acetylated or tyrosinated, because our results demonstrate that many microtubule segments and cross-sections are either mostly acetylated or mostly tyrosinated. Nonetheless, we also report a considerable amount of chemical crosstalk, measured as ↑ and →. Furthermore, since our current methodology does not allow us to trace entire microtubules, it is still possible that microtubules feature segments with different modifications. We now mention this more explicitly in the discussion.

4) Along the lines of point 1) and 2), the authors might consider discussing the limitation of the method to quantify PTMs that are intrinsically rheostatic, such as polyglutamylation.

We thank the reviewer for this suggestion and have added this to the discussion.

5) The authors describe that filaments positive for acetylation show a decreased intensity for an α-tubulin antibody, interpreting that this might be due to decreased efficiency of the staining on this highly modified microtubule. A similar clash in compatibility might occur for acetylation and tyrosination staining (considering that both stainings are observed to different extent in both microtubule subsets). Can they exclude that this happens?

It could indeed be possible that the staining for tyrosinated tubulin is weaker on highly acetylated microtubules, while the actual level of tyrosination is similar to non-acetylated microtubules. Nonetheless, although technical problems precluded us from testing this directly in neurons, in non-neuronal cells acetylation and detyrosination clearly overlap (Figure 1—figure supplement 4A). Furthermore, even if we underestimate the level of tyrosination on acetylated microtubules or the level of acetylation on tyrosinated microtubules, the fact that we clearly observe two populations of microtubules must reflect a true chemical differentiation between these microtubules.

6) Following point 1) and 4), it would be a great addition to the current paper if the authors could show that they can detect the non-tyrosinated microtubules with an antibody for detyrosination. As it is likely that the authors have already tried this, another alternative could be delta2-tubulin. In neurons, long-lived microtubules often end up being delta2-positive following detyrosination. If none of this worked in the hands of the authors, they should still mention in the manuscript that the co-staining of the acetylated microtubule population with either of these two antibodies would be expected. This discussion could also highlight the limitations of the currently available antibodies to tubulin PTMs.

As indicated in point 2, we have now performed additional experiments using different antibodies.

7) When imaging the dendrites, the authors select a region of length 5-10 μm proximal to the soma. They find that the ratios of mostly tyrosinated/mostly acetylated microtubules are very similar in both compartments. It would be informative to gain insight into the subsets of modified microtubules along the entire length of a dendrite, or at least at multiple sections closer to the proximal end. In Figure 1D and 4C the intensity distributions for tyrosination and acetylation suggest that shifts in the subsets distribution might occur further down the dendrite.

In the revised manuscript, we include a whole-dendrite dataset for which we were able to measure the microtubule intensity as a function of dendrite thickness (Figure 1—figure supplement 3C). We found that microtubule density displayed a quadratic dependence on dendrite diameter over a wide range of diameters (i.e. proportional to cross-section area). Only for the very thin dendrites (FWHM < 1µm) we observed a deviation from this dependence. These results indicate that microtubule density remains fairly constant throughout the dendrite. These results also suggested that the intensity of acetylation and tyrosination drops slightly more than overall tubulin intensity, which could reflect the emergence of a new subset of non-acetylated and non-tyrosinated microtubules. However, given the limitations of this analysis, we prefer to not draw too strong conclusions from these data.

8) The newly developed FlipExM approach improves the resolution of single microtubules significantly, however, the method has the limitation of observing only a single cross section (1 cm x 3 mm) of the dendrite, on which the conclusions are based. The authors should explain how the specific regions of these cross sections were selected and controlled for the six different cells used for replicating the experiments (for example: which distance from the cell body was chosen?).

We focused on the detailed analysis of the proximal dendrites for various reasons. First, neuronal cultures are very dense and thin neurites from different cells often overlap and intertwine, which makes it challenging to separate and trace individual neurites further away from the cell body. In addition, when performing FlipExM, we use the cell bodies at the glass-gel interface as a reference point to identify dendrites. Since these dendrites are now aligned with the z-axis, this limits the dendritic length that we can image. We used the first 5-10 µm, which is consistent with the data shown in Figures 2 and 3.

9) From Figure 5H and to some extent from Figure 3C it seems that with increasing diameter of the dendrite, and respectively microtubule number, the fraction of the acetylated filaments increases proportionally, while the peripheral tyrosinated and others fractions remain roughly the same. This may have an influence on the absolute numbers quantified and reported in this study about the percentage of the investigated modifications, and should be discussed.

The new data set where we analyze an entire dendrite using STED shows that the relative change in acetylation and tyrosination levels along the dendrite is very comparable (Figure 1—figure supplement 3A,B)

10) The supplementary videos are very well done, and they help the reader to easily visualise the described approaches. Could the authors imagine to add a voice chancel explaining to the reader what is happening on the video? Or alternatively, make the explanatory text appear on the videos?

Following the reviewer’s suggestion, we first tried adding narration to the video. Because the results were unsatisfactory, we then chose for having explanatory text appear on the video.